# A Theory of Initialisation's Impact on Specialisation

**Devon Jarvis**[*,1,5], **Sebastian Lee**[*,2], **Clémentine Carla Juliette Dominé**[*,3],
**Andrew M Saxe**, [3,6] **& Stefano Sarao Mannelli**[4,1]

## Abstract

Prior work has demonstrated a consistent tendency in neural networks engaged in continual learning tasks, wherein intermediate task similarity results in the highest levels of catastrophic interference. This phenomenon is attributed to the network's tendency to reuse learned features across tasks. However, this explanation heavily relies on the premise that neuron specialisation occurs, i.e. the emergence of localised representations. Our investigation challenges the validity of this assumption. Using theoretical frameworks for the analysis of neural networks, we show a strong dependence of specialisation on the initial condition. More precisely, we show that weight imbalance and high weight entropy can favour specialised solutions. We then apply these insights in the context of continual learning, first showing the emergence of a monotonic relation between task-similarity and forgetting in non-specialised networks. Finally, we show that specialization by weight imbalance is beneficial on the commonly employed elastic weight consolidation regularisation technique.

## 1 Introduction

Theories of representation in biological neural networks span from highly localised representations in single neural units (Barlow, 1972) to fully distributed or shared representations (Hopfield, 1982). While shared representations offer greater resilience, specialised representations allow for more efficient encoding of information. Experimental evidence supports both ends of this spectrum, with different brain areas and tasks exhibiting distinct forms of representation (Blakemore et al., 1973; Quiroga et al., 2005; Georgopoulos et al., 1986; Ishai et al., 2000; Averbeck et al., 2006). Similarly, artificial neural networks display both shared (LeCun et al., 1989; Erhan et al., 2010; Yosinski et al., 2014) and specialised representations (Zeiler & Fergus, 2014; Voita et al., 2019), where a recent advancements in explainable AI, such as the Golden Gate Claude model (Templeton, 2024), exemplify an extreme of the spectrum.

Given the trade-off between shared and specialised representations, a critical research challenge lies in understanding how to guide neural networks towards one form or the other. This tension is especially relevant in contexts like disentangled representation learning (Bengio et al., 2013) and multi-task learning (Caruana, 1997), including continual learning and transfer learning. Specialised representations can facilitate faster adaptation and reduce catastrophic forgetting (McCloskey & Cohen, 1989; Ratcliff, 1990), as they allow networks to rewire efficiently (Suddarth & Kergosien, 1990). Rich Caruana's seminal work on multi-task learning (Caruana, 1997) emphasised the value of specialisation in enhancing performance across multiple tasks. In disentangled representation learning, Locatello et al. (2019) highlighted that, despite the potential success of unsupervised approaches, disentanglement does not emerge naturally without an explicit inductive bias, underscoring the need for supervision or regularisation to enforce such structures. Thus, recent efforts to mitigate catastrophic forgetting (Parisi et al., 2019; De Lange et al., 2021) have led to the development of regularisation strategies that promote specialisation, such as elastic weight

[1]School of Computer Science and Applied Mathematics, University of the Witwatersrand; [2]Center for Computational Neuroscience, Flatiron Institute, Simons Foundation; [3]Gatsby Computational Neuroscience Unit & Sainsbury Wellcome Centre, UCL; [4]Data Science and AI, Computer Science and Engineering, Chalmers University of Technology and University of Gothenburg; [5]Machine Intelligence and Neural Discovery Institute, University of the Witwatersrand; [6]CIFAR Azrieli Global Scholar, CIFAR;
*Equal contribution in random order.

consolidation (Kirkpatrick et al., 2017), synaptic intelligence (Zenke et al., 2017), and learning without forgetting (Li & Hoiem, 2017).

This paper aims to show that initialisation has fundamental importance in achieving specialised solutions, providing a complementary perspective on both the lazy (Jacot et al., 2018) and rich learning regimes (Mei et al., 2018; Chizat & Bach, 2018; Rotskoff & Vanden-Eijnden, 2018). Previous research (Chizat et al., 2019; Geiger et al., 2020; Bordelon & Pehlevan, 2022) has showns that by interpolating between these regimes, we can transition from shared representations–characterised by random projections in the neural tangent kernels–to effective feature learning (Tarmoun et al., 2021; Kunin et al., 2024; Xu & Ziyin, 2024; Dominé et al., 2024; Varre et al., 2024). While our analysis remains within the feature learning regime, it adopts a distinct theoretical approach compared to these studies, concentrating specifically on the impact of initialisation within standard synthetic frameworks for neural networks. This exploration reveals how initialisation can skew the learning dynamics towards either specialised or shared representations, thereby adding a new dimension to the study of learning dynamics in over-parameterised networks.

Our work makes the following **main contributions**:

- We study the impact of initialisation on specialisation through two theoretical frameworks:
  - We utilise the dynamics of **deep linear networks** to investigate the evolution of specialisation (Saxe et al., 2013);
  - We extend this analysis to **high-dimensional mean-field neural networks** learning with stochastic gradient descent (Saad & Solla, 1995b;a; Biehl & Schwarze, 1995).
- We identify specific initialisation schemes that promote specialised solutions by increasing the entropy of the readout weights and creating an imbalance between the first and last layers, akin to the findings of Dominé et al. (2024).
- We demonstrate that there are two regimes of forgetting profiles contingent on neuron specialisation, reconciling recent findings regarding the non-monotonic relationship between task similarity and catastrophic forgetting (Ramasesh et al., 2020; Lee et al., 2021; 2022) with the traditional belief in monotonic forgetting (Goodfellow et al., 2013).
- Finally, we demonstrate the practical implications of our results on regularisation strategies, specifically analysing how Elastic Weight Consolidation (EWC) (Kirkpatrick et al., 2017) is influenced by specialisation dynamics, highlighting potential pitfalls associated with regularisation methods in continual learning.

Fig. 1 of Sec. 2 serves as a motivating figure and contrasts with results from the literature (Goldt et al., 2019) that sigmoidal networks specialise while ReLU networks do not. We show that ReLU networks can also specialise (and sigmoidal networks do not always specialise) depending on weight initialisations. Sec. 3 then aims to understand in more detail "what" aspects of the initialisation scheme lead to specialised solutions. This is achieved theoretically through the lens of deep linear network dynamics (Saxe et al., 2013) and validated empirically on a canonical disentanglement task with a variational autoencoder (VAE) in Sec. 3.2. Having established an initialisation strategy which promotes specialisation, we then apply this strategy to control specialisation when studying continual learning in Sec. 4. Consequently, we reconcile the two distinct forgetting profiles observed in practice which determine how an increase in task similarity leads to forgetting of earlier tasks: 1. the Monotonic profile (Goodfellow et al., 2013), 2. the Maslow's Hammer profile (Ramasesh et al., 2020; Lee et al., 2021; 2022). We demonstrate that the Maslow's Hammer profile results from neuron specialisation, incurs less interference as tasks become significantly different and enables regularisation techniques like Elastic Weight Consolidation (EWC) Kirkpatrick et al. (2017). Finally, in Sec. 5, we reflect on the limitations of our work and propose future directions for research.

## 2 SPECIALISATION IN THE TEACHER-STUDENT FRAMEWORK

The teacher-student framework is a generative model that allows for the controlled creation of synthetic datasets (Gardner & Derrida, 1989). The framework involves two classifiers: the *teacher* and the *student*, for instance represented as neural networks as exemplified in Fig. 1a. The teacher, has fixed randomly drawn weights and maps random inputs $x$ from a given distribution to labels, providing a rule for generating data. The student, on the other hand, updates its parameters through learning protocols like stochastic gradient descent (SGD) to approximate the teacher's outputs.

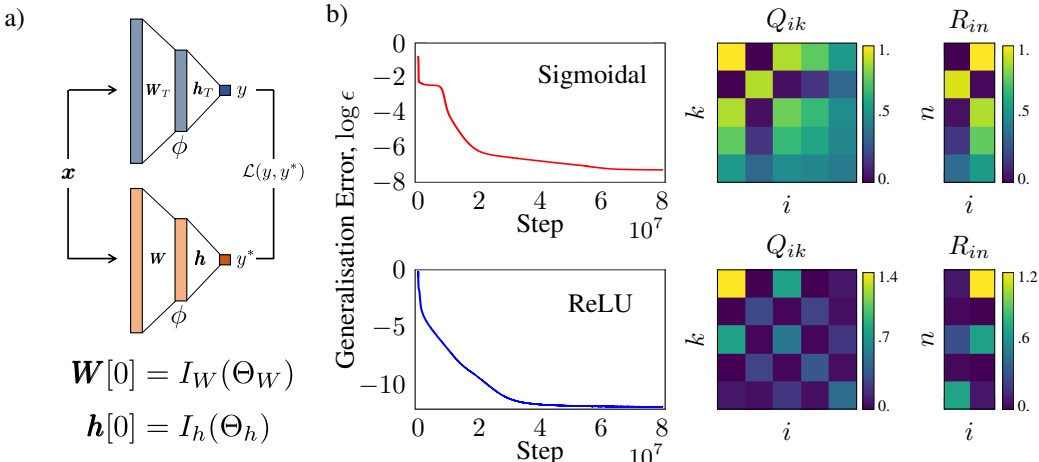

Figure 1: **Initialisation impacts specialisation.** a) In the teacher-student setup a student network is trained with labels generated by a fixed teacher network. Previous work established a relationship between the activation function $\phi$ and the propensity for the student nodes to specialise to teacher nodes. However we show in this work that this is an overly simplistic description; other factors including student weight initialisations $I_W, I_h$, parameterised by $\Theta_W, \Theta_h$ arguably play a stronger role. b) Generalisation error curves for two simulations of the teacher-student setup, one with a ReLU activation function and one with a scaled error activation function. $\Theta_W$ and $\Theta_h$ are chosen to achieve a solution with ReLU that specialises—as indicated by sparser overlap matrices on the bottom right, and a scaled error function solution that does not specialise—as indicated by denser overlap matrices on the top right. A sparse (dense) $Q$ matrix shows few (many) student nodes are active, while a sparse (dense) $R$ matrix shows student nodes are representing teacher nodes in a targeted (redundant) manner. Further details for the quantities described can be found in Sec. 4.

While a detailed quantitative characterisation of specialisation follows in the next sections, we briefly introduce the concept within the teacher-student framework. Saad & Solla (1995b) showed that, when both teacher and student are modelled as committee machines, each student neuron specialises by aligning with a specific teacher neuron. Similarly, Goldt et al. (2019) observed that for certain activation functions in two-layer networks, an over-parameterised student will selectively use only a subset of those units to replicate the teacher's outputs. This phenomenon, termed specialisation, stands in contrast to a student redundantly sharing representations of the teacher across neurons. In this work we present a more comprehensive account of the factors underlying specialisation. In contrast to (Goldt et al., 2019), we argue that initialisation—not the activation function—is chiefly responsible. We highlight this in Fig. 1b, by showing that with carefully chosen initialisations we can train a highly specialised ReLU student (bottom panels), and a non-specialising sigmoidal student (top panels)–shown by the sparser $Q$ and $R$ matrices of the ReLU network–which represents the opposite of the conclusions presented in (Goldt et al., 2019). We begin by aiming to establish what properties of an initialisation promote specialisation. This question is well suited to the deep linear network theory (Saxe et al., 2013) and we turn to this strategy now.

## 3 SPECIALISATION EXPLAINED USING LINEAR DYNAMICS

Here we construct a synthetic setup, to study the influence of initialisation on specialisation using the deep linear network framework (Saxe et al., 2022; 2019). While deep linear networks can only represent linear input-output mappings, they showcase intricate fixed point structure and nonlinear learning dynamics reminiscent of phenomena seen in nonlinear networks (Baldi & Hornik, 1989; Fukumizu, 1998; Arora et al., 2018; Lampinen & Ganguli, 2019). We consider specialisation adhering to the definition proposed by the statistical physics literature (Goldt et al., 2019) which considers whether one neuron will account for all of the variance associated to one feature, while the others remain inactive (a phenomenon close to activation sparsity and reminiscent of how initial conditions can lead to minimal subnetworks in the "Lottery Ticket Hypothesis" (Frankle & Carbin, 2018)).

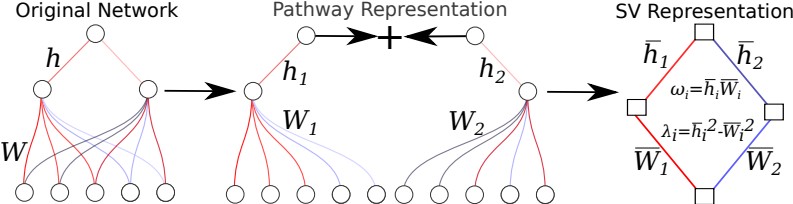

Figure 2: Summary of our setup, notation and strategy. a) The original network with two hidden neurons learning the regression task. b) We split the network into two separate pathways and consider their dynamics individually. Since both networks are learning the same task simultaneously, their dynamics are coupled. c) To obtain the dynamics of the two pathways and calculate their escaping and hitting time we track the pathway dynamics in terms of the network's effective singular values. The closed form dynamics for the pathway singular value are given in Eq. 3.

This is in contrast to other work on modularity (Jarvis et al., 2023) such as Neural Module Networks (Andreas et al., 2016; Hu et al., 2017; 2018; Andreas, 2018), mixture-of-expert models (Masoudnia & Ebrahimpour, 2014; Bengio et al., 2015; Shazeer et al., 2017), tensor product networks (Smolensky et al., 2022), among others (Chang et al., 2018; Goyal et al., 2019), which consider specialisation as a subset of a network or module performing a single "task" or only being activated by one interpretable feature in the dataset. Thus, these works are focusing on specialisation to imply feature sparsity (Dasgupta et al., 2022), while we are concerned with the learning mechanism that leads to sparse input-output mappings.

### 3.1 SPECIALISATION IN THE DEEP LINEAR NETWORK FRAMEWORK

To connect this framework to specialisation we use the notion of the "neural race" from Saxe et al. (2022). The neural race hypothesis says that the pathways through a network are racing to explain the variance in the dataset (i.e. to perform the input-output mapping). Thus, we consider the limited case of a network with two hidden neurons and one output neuron. Fig. 2 depicts the setup, notation and strategy for this section. By defining "hitting time" as how long it takes a pathway to reach its final converged value, and "escaping time" as how long it takes the pathway to begin learning, we ask the question: "*when will one pathway finish learning (reach it's hitting time $t^*$) before the other begins learning (reaches it's escaping time $\hat{t}$)*". In cases when this occurs, the network would have specialised as only one pathway will have any activity and will explain all of the data. Similar to Sec. 4 we generate data by sampling the elements of a data point from a Gaussian distribution ($x_i \sim \mathcal{N}(0,1)$) with $i = 1, \ldots, d$. We then define a ground-truth mapping ($\boldsymbol{W}_T$) and generate labels $y = \boldsymbol{W}_T \cdot \boldsymbol{x}$. We only consider regression tasks in this section, thus $y \in \mathbb{R}$. For $n$ inputs we can form the input matrix $\boldsymbol{X} \in \mathbb{R}^{d \times n}$ and row vector of scalar outputs $\boldsymbol{y} \in \mathbb{R}^{1 \times n}$. The dataset statistics which drive learning are collected in the input and input-output correlation matrices, $\Sigma^x$ and $\Sigma^{yx}$ respectively. For the task described above the singular value decomposition of these matrices are:

$$\boldsymbol{\Sigma}^x = E[\boldsymbol{X}\boldsymbol{X}^T] = \boldsymbol{V}\boldsymbol{D}\boldsymbol{V}^T, \qquad \boldsymbol{\Sigma}^{yx} = E[\boldsymbol{y}\boldsymbol{X}^T] = u s \boldsymbol{v}^T. \qquad (1)$$

Here, $u \in \{-1, 1\}$, $\boldsymbol{v}$ is a vector such that $\boldsymbol{v}^T\boldsymbol{v} = 1$ and $\boldsymbol{V}$ is an orthogonal singular vector matrix. Correspondingly, $s$ is the singular value for the rank 1 task and $\boldsymbol{D}$ is a diagonal matrix of singular values. Note that – as in Saxe et al. (2013; 2022) – we assume that the correlation matrices are mutually diagonalisable (share the same $\boldsymbol{V}$) up to the rank of $\Sigma^{yx}$.

For this task we consider a single hidden layer network (Fig. 2 left) computing output $\hat{y} = \boldsymbol{h}\boldsymbol{W}\boldsymbol{x}$ with $\boldsymbol{h} \in \mathbb{R}^p$ and $\boldsymbol{W} \in \mathbb{R}^{p \times d}$ in response to an input $\boldsymbol{x} \in \mathbb{R}^d$. The network is trained to minimise the mean squared error loss using full batch gradient descent with a small learning rate $\eta$. To identify when specialisation will occur in this network, we split the network into two pathways with one hidden neuron each. The input and output dimensions remain the same (Fig. 2 middle). Finally we obtain the linear dynamics (ultimately depicted as Eq. 3) for each pathway (the full details and assumptions of the derivation are given in Appendix A). In this setting, the pathway's input-output mapping after $t$ epochs of training is $h(t)\boldsymbol{w}(t)$. Notice that, the pathways have one hidden neuron and so $h$ is a scalar and $w$ is the vector of input to respective hidden neuron weights, to once again alleviate notation we do not denote which pathway. Assuming that the pathway weights align to the singular vectors of the dataset from early in training, as described by the "*silent alignment effect*" (Atanasov et al., 2021),

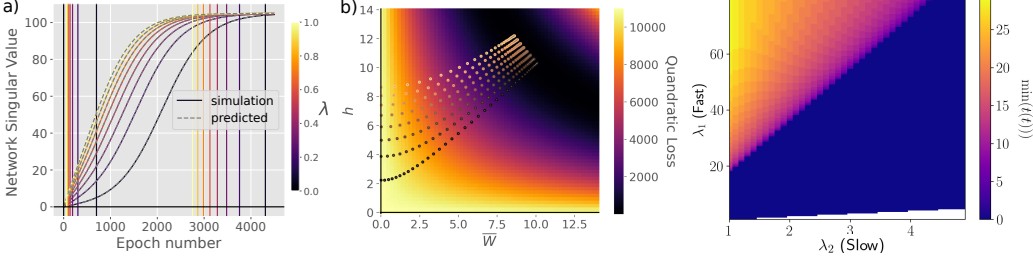

Figure 3: **Linear Dynamics from imbalanced initialisation leads to specialisation.** *Panels a-b)* Show agreement between our theoretical curves and simulations for the training dynamics of: (a) the network's singular value dynamics, escaping times (verticals towards left) and hitting times (verticals towards right) for varying scales of weight imbalance $\lambda$ (depicted by colour), (b) and the network's movement in weight space depicted by the sequence of dots over weight space. Colour depicts the loss of the network configuration at a point. *Panel c)* shows a phase diagram representing how pathways with different initial weight imbalances lead to specialisation. The two axis represent the weight imbalance of the two pathways in our broader network ($\lambda_2$ on the x-axis for the slower pathway and $\lambda_1$ on the y-axis for the faster pathway). The colour represents how close the slower pathway is to reaching its escaping time at its closest point throughout training (in $\log$ scale). We see that the more inbalanced the fast pathway relative to the slower pathway, the more likely the network will specialise. The white region represents when the inbalance is equal or reversed.

we perform a change of variables and write the mapping in terms of the dataset singular vectors:

$$h(t)\boldsymbol{w}(t) = u\omega(t)\boldsymbol{v}^T, \qquad (2)$$

where $\omega(t)$ is the pathway's scalar effective singular value and the only time-dependent component of the decomposed mapping. While the alignment assumption is strong, linear paradigms with these assumptions have been used successfully in the past (Saxe et al., 2019; Lampinen & Ganguli, 2019; Braun et al., 2022; Jarvis et al., 2023; Dominé et al., 2024; Jarvis et al., 2025). By appropriately changing variables, we can obtain a closed form equation describing how $\omega$ evolves through time as:

$$\omega(t) = \frac{\lambda}{2}\sinh\left\{2\tanh^{-1}\left[\frac{k\left(c\exp\left(\frac{sgn(\lambda)k}{\tau}t\right) - 1\right) - \lambda d\left(c\exp\left(\frac{sgn(\lambda)k}{\tau}t\right) + 1\right)}{2s\left(c\exp\left(\frac{sgn(\lambda)k}{\tau}t\right) + 1\right)}\right]\right\} \qquad (3)$$

where $c$ is a defined constant, $\tau = \frac{1}{\eta}$ is the learning time constant and $k = \sqrt{4s^2 + \lambda^2 d^2}$. Eq. 3 shows that $k$ is the variable interacting with time ($t$) and as a consequence determines how quickly the network will learn. Three factors affect $k$ fastening learning: 1. $s$ the input-output correlation matrix singular value, 2. $d$ the input correlation matrix singular value, and 3. $\lambda = h^2 - \boldsymbol{ww}^T$ which denotes the imbalance between the weights of the network. Notice that–as shown in Appendix A–$\lambda$ is a conserved quantity and constant throughout training. Thus, given a dataset–which determine the $s$ and $d$ matrices–the only property which can promote faster learning in the network is to increase the imbalance parameter. For our experiments we whiten the input data $\boldsymbol{x}$ such that $k = \sqrt{4s^2 + \lambda^2}$ to remove one of the interactions within $k$. With the training dynamics of a singular value defined as in Eq. 3, we can formally define the escaping time as $\hat{t} = t$ such that $\omega(t) = \delta$ for a small $\delta \in \mathbb{R}$. Similarly, we define the hitting time as $t^* = t$ such that $\omega(\infty) - \omega(t) = \delta$ for a small $\delta \in \mathbb{R}$.

Fig. 3(a-b) show a confirmation of the validity our theory by comparing with simulations. Instead, Fig. 3c represents the main result of this section. We consider both network pathways and vary the weight imbalance for each ($\lambda_{slow}$ for the pathway with the lower imbalance and $\lambda_{high}$ for the pathway with the larger imbalance). We place these two values on the axes and in colour depict how close the slower pathway comes to reaching its escaping time across its training ($\min(\hat{t}_{slow}(t))$). When zero, it means that during training there is a timestep where the network is less than one epoch from its escaping time (so it will learn). In this case there will not be specialisation as both pathways will learn some part of the input-output mapping. When the colour is positive it means there will be specialisation as the slower pathway is always at least a full epoch away from learning. It is important to note that the slower pathway's escaping time is moving constantly as the faster pathway accounts for variance in the data. This decreases the input-output singular value in $k$ for this pathway and makes learning slower. Due to this coupling we are also unable to obtain completely closed form

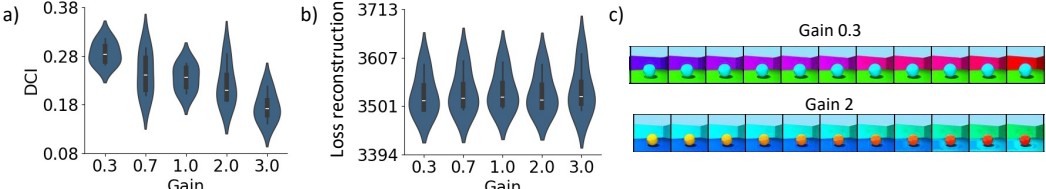

Figure 4: Violin plots of *a)* the Disentanglement, Completeness, and Informativenes (DCI) East-wood & Williams (2018) score and *b)* the reconstruction loss against gain. The disentanglement score decreases as the gain increases while the reconstruction loss remains steady, *c)* Example traversals of models with gains 2 and 0.3, respectively, highlighting a disentangled dimension for gain 0.3 and a mixed dimension for gain 2. Experimental details can be found in appendix D.

equations for the slower pathway in term's of the faster pathway's effective singular value. However, this phase diagram would not be computationally feasible without the closed-form escaping time, hitting time and training dynamics (see Appendix B for our process on constructing this plot). Finally, we only consider imbalances where the output scale is larger than the input scale. Recent work (Kunin et al., 2024; Dominé et al., 2024) has shown that having larger input weight scale pushes the network towards lazy learning while output heavy imbalance promotes feature learning. From Fig. 3 we see that there is a clear phase transition from non-specialised representations to specialised ones. This occurs with increasing imbalance of the faster pathway. Increasing the imbalance of the slower pathway can similarly combat this specialisation pressure. Thus, the relative imbalance of the two pathway at initialisation will dictate whether specialised representations are learned.

## 3.2 INCREASING SPECIALISATION IN DISENTANGLED REPRESENTATION LEARNING

To empirically support the linear network theory, we extend the results on inbalanced initialisation and apply them, beyond the limited setting of our framework, in the context of disentangled representation learning, where the goal is to separate latent factors into both feature and activity sparse neural representations. For further empirical support of our theoretical results on Sparse Autoencoders (SAEs) see App. H. Bengio et al. (2013) introduced the importance of disentanglement for interpretability and generalisation. A seminal contribution to this domain came with the $\beta$-VAE model, where Higgins et al. (2017) demonstrated how increasing the KL-divergence term can enforce disentanglement by encouraging specialised latent representations. Many studies have built upon these foundational frameworks to enhance disentanglement performance, exploring different training regimes (Locatello et al., 2020; Fumero et al., 2021) and loss functions (Chen et al., 2019; Kim & Mnih, 2019; Kumar et al., 2018). Here we contribute to this literature by applying our theoretical insights and examining the impact of initialisation on disentanglement performance.

Specifically, we examine how initialisation impacts specialisation in disentanglement learning on the 3DShapes dataset (Burgess & Kim, 2018) using the $\beta$-VAE model–widely adopted for such tasks (Higgins et al., 2017; Burgess et al., 2018). We implement a $\beta$-VAE model, employing the "Deep-GaussianLinear" architecture for the decoder and the "DeepLinear" architecture for the encoder, as specified in Locatello et al. (2019). Both architectures are composed of five fully connected layers with ReLU activations. The model is trained using the Adam optimiser, optimising a loss function that combines KL divergence and binary cross-entropy-based reconstruction loss. Additional details are given in Appendix D. In these experiments, we adjust the variance of the weights in a deep fully-connected encoder, by varying the constant gain of the Xavier initialisation (Glorot & Bengio, 2010). Specifically, the first block of layers was initialised with gain $g$ while the readout layer received a gain $1/g$. Notice that $g = 1$ represents the standard initialisation scheme.

Results are shown in Fig. 4, despite very similar levels of reconstruction loss, networks initialised with smaller gains improved disentanglement in the $\beta$-VAE network, as reflected in higher Disentanglement, Completeness, and Informativeness (DCI) scores (Eastwood & Williams, 2018). This result confirms that modulating the initialisation gain can increase the network's disentanglement. Although the scope of these experiments is limited, they provide preliminary validation of our theoretical framework in more realistic contexts, encouraging further investigation into alternative initialisa-

tion schemes with varying levels of balance. Having investigated the role initialisation plays in promoting specialisation, we return to the original setting of Sec. 2 to understand the role of initialisation in governing network behaviour during continual learning. In the light of these results we aim to revisit the two established forgetting profiles empirically observed in the continual learning literature: Namely, the Maslow's Hammer profile, observed empirically first in Ramasesh et al. (2020), and the monotonic forgetting profile, more typically assumed and observed in Goodfellow et al. (2013).

## 4    CONTINUAL LEARNING

As Caruana (1997) noted, multi-task learning benefits significantly from task-specific specialisation, allowing the network to better preserve performance across multiple domains. In the context of continual learning, Ramasesh et al. (2020) and Lee et al. (2021) observed that forgetting does not monotonically increase with task similarity. Lee et al. (2022) provided a mechanistic explanation, showing that this phenomenon is due to the interplay between re-use of specialised neurons and activation of unused ones. In this section, we build on these findings and show that this phenomenology can be disrupted by initialisation schemes that disincentives specialisation.

### 4.1    CONTINUAL LEARNING IN THE TWO-LAYER TEACHER-STUDENT SETUP

We use a teacher-student framework, introduced in Sec. 2, which has been analysed in Lee et al. (2021; 2022). This model consists of two randomly initialised teacher networks—one for an upstream task and one for a downstream task. Each teacher is represented by two-layer neural networks with $p^*$ hidden units and weights $\boldsymbol{W}_T^{(1)}, \boldsymbol{h}_T^{(1)}$ for the upstream task, and $\boldsymbol{W}_T^{(2)}, \boldsymbol{h}_T^{(2)}$ for the downstream task. Given a random input $\boldsymbol{x} \in \mathbb{R}^d$, drawn i.i.d. from a Gaussian distribution $x_i \sim \mathcal{N}(0, 1)$, the teachers generate labels according to the equation:

$$y^{(t)} = \boldsymbol{h}_T^{(t)} \cdot \phi\left(\frac{\boldsymbol{W}_T^{(t)}\boldsymbol{x}}{\sqrt{d}}\right) \quad \text{for } t = 1, 2, \tag{4}$$

where $\phi$ is a non-linear activation function, chosen here as $\phi(z) = \text{erf}\left(z/\sqrt{2}\right)$. This setup allows us to generate two datasets $\mathcal{D}^{(1)}$ and $\mathcal{D}^{(2)}$, with controlled similarity between the tasks by manipulating the teacher weights. Specifically, we generate $\boldsymbol{W}_T^{(1)}, \boldsymbol{h}_T^{(1)}$, and $\boldsymbol{h}_T^{(2)}$ with i.i.d. Gaussian entries, while $\boldsymbol{W}_T^{(2)}$ is generated as:

$$\boldsymbol{W}_T^{(2)} = \gamma\boldsymbol{W}_T^{(1)} + \sqrt{1 - \gamma^2}\boldsymbol{W}_T^{(\text{aux})}, \tag{5}$$

where $\boldsymbol{W}_T^{(\text{aux})}$ is an auxiliary weight matrix, and $\gamma$ controls the correlation between tasks. The student is a two-layer neural network with $p$ hidden units, using the same non-linearity $\phi$. It is trained using online stochastic gradient descent on a squared error loss, with a shared first-layer weight matrix $\boldsymbol{W}$ and task-specific readout weights $\boldsymbol{h}^{(1)}$ and $\boldsymbol{h}^{(2)}$. For both layers, the initial weights are sampled i.i.d. from a Gaussian distribution, with the first-layer weights $\boldsymbol{W}$ having standard deviation $\sigma_W$. While most previous studies follow a similar scheme for the readout weights, we introduce a novel initialisation scheme using polar coordinates. The updates for $\boldsymbol{W}$ and $\boldsymbol{h}^{(t)}$ at iteration $e$, under SGD on the squared error loss, are given by:

$$\boldsymbol{W}[e + 1] = \boldsymbol{W}[e] - \frac{\eta}{\sqrt{d}}\left(\boldsymbol{h}^{(t)} \cdot \phi\left(\frac{\boldsymbol{W}\boldsymbol{x}}{\sqrt{d}}\right) - y^{(t)}\right)\phi'\left(\frac{\boldsymbol{W}\boldsymbol{x}}{\sqrt{d}}\right)\boldsymbol{v}^{(t)}\boldsymbol{x}, \tag{6}$$

$$\boldsymbol{h}^{(t)}[e + 1] = \boldsymbol{h}^{(t)}[e] - \frac{\eta}{d}\left(\boldsymbol{h}^{(t)} \cdot \phi\left(\frac{\boldsymbol{W}\boldsymbol{x}}{\sqrt{d}}\right) - y^{(t)}\right)\phi\left(\frac{\boldsymbol{W}\boldsymbol{x}}{\sqrt{d}}\right), \tag{7}$$

where $\eta$ is the learning rate and $y^{(t)}$ is the target output from the teacher network for task $t$. In the large input dimension limit $d \rightarrow \infty$, key observables, such as the generalisation error, can be captured by a few order parameters:

$$\boldsymbol{Q} = \frac{1}{d}\boldsymbol{W}\boldsymbol{W}^T, \qquad \boldsymbol{R}^{(t)} = \frac{1}{d}\boldsymbol{W}\boldsymbol{W}_T^{(t),T}, \qquad \boldsymbol{T}^{(t,t')} = \frac{1}{d}\boldsymbol{W}_T^{(t)}\boldsymbol{W}_T^{(t'),T}, \qquad \boldsymbol{h}^{(t)}, \qquad \boldsymbol{h}_T^{(t)}; \tag{8}$$

where $t, t' \in \{1, 2\}$ refer to the two tasks. The generalisation error for task $t$ is then:

$$\epsilon^{(t)} = I_{21}(\boldsymbol{Q}, \boldsymbol{h}^{(t)}) + I_{21}(\boldsymbol{T}^{(t,t)}, \boldsymbol{h}_T^{(t)}) - \frac{1}{2}I_{22}(\boldsymbol{Q}, \boldsymbol{R}^{(t)}, \boldsymbol{T}^{(t,t)}, \boldsymbol{h}^{(t)}, , \boldsymbol{h}_T^{(t)}), \tag{9}$$

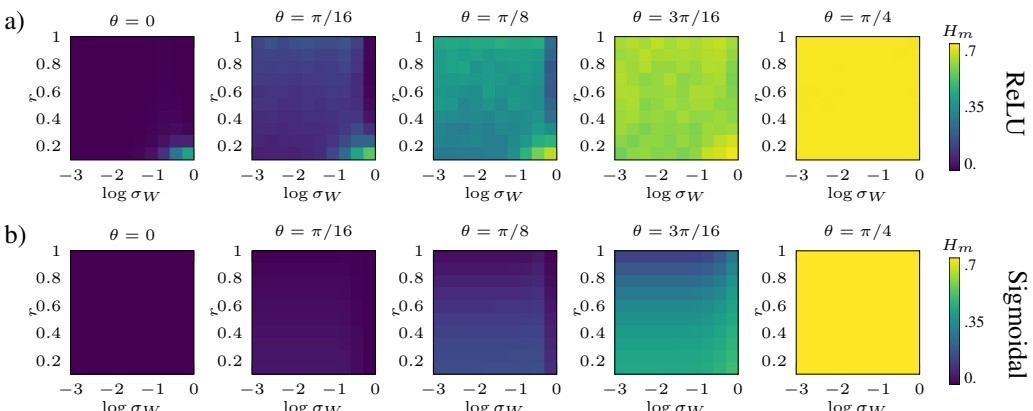

Figure 5: **Phase diagrams show significance of initialisation for specialisation.** The phase diagrams show with colour the aggregated entropy Eq. 10 evaluated for different initialisations. On the x-axis we span over the standard deviation of the first layer. The second layer is initialised using polar coordinates, and the y-axis represents the norm while the different panels give the angle spanning from orthogonal units ($\theta = 0$) to identical units ($\theta = \pi/4$). Specialisation is achieved by blue-leaning initialisations, while yellow-leaning ones exhibit high entropy and therefore non-specialised solutions. Additional results can be found in Appendix E.

where $I_{21}$ and $I_{22}$ are explicit functions of the order parameters, detailed in Appendix C. The evolution of these parameters throughout training can be tracked to study the learning dynamics, as first shown in Saad & Solla (1995a); Biehl & Schwarze (1995); Goldt et al. (2019). For the specific case of continual learning, Lee et al. (2021) derived the governing ordinary differential equations (ODEs), provided in Appendix C.

## 4.2 SPECIALISATION'S RELEVANCE FOR CONTINUAL LEARNING

The continual learning results in the teacher-student setup, including the non-monotonic relationship between catastrophic forgetting and task similarity, often implicitly assume that the student has specialised to the teacher in the first task. This assumption allows for spare capacity to represent the second task. However, as shown in Fig. 1b, there are regimes where this assumption of specialisation is violated. Here, we expand on these findings and their implications for forgetting.

A student can effectively ignore a unit in two ways: either the unit's post-activation is near 0 (inactive), or the corresponding second-layer weight is 0. This motivates three measures for specialisation based on the definition of entropy–over the hidden units, head weights, and the product of both:

$$H_h = -\sum_i^p |\tilde{h}_i| \log |\tilde{h}_i|, \quad H_Q = -\sum_i^p \tilde{Q}_{ii} \log \tilde{Q}_{ii}, \quad H_m = -\sum_i^p \tilde{Q}_{ii} |\tilde{h}_i| \log(\tilde{Q}_{ii} |\tilde{h}_i|); \quad (10)$$

where the tilde denote normalisation, i.e. $|\tilde{h}_i| = \frac{|h_i|}{\sum_i^P |h_i|}$ and $\tilde{Q}_{ii} = \frac{Q_{ii}}{\sum_i^P Q_{ii}}$. Maximum entropy corresponds to no specialisation, while minimum entropy corresponds to maximum specialisation.

We can investigate how these measures vary as a function of different properties of the problem setup, in particular those related to initialisation. To simplify the analysis, we begin with the case where the optimal number of tasks is $p^* = 1$ and the network has $p = 2$ output units. This allows us to initialise the second layer weights in polar coordinates, with precise and interpretable control over scale and asymmetry of weights. Formally we parameterise our readout initialisations according to $\boldsymbol{h}^{(t)}[0; r^{(t)}, \theta^{(t)}] = (r^{(t)} \cos \theta^{(t)}, r^{(t)} \sin \theta^{(t)})$. Fig. 5 contain phase diagrams showing how the entropy measures in Eq. 10 vary with the initialisation parameters $r^{(t)}, \theta^{(t)}$, and $\sigma_W$. We can make several observations: (i) the strongest determinant of specialisation is the asymmetry in the second layer weights, i.e. the $\theta$ parameter. (ii) this is the case for both ReLU and sigmoidal activation functions, reinforcing the point made in the example from Fig. 1b. (iii) the scale of initialisations (parameters $\sigma_W, r$) are also important.

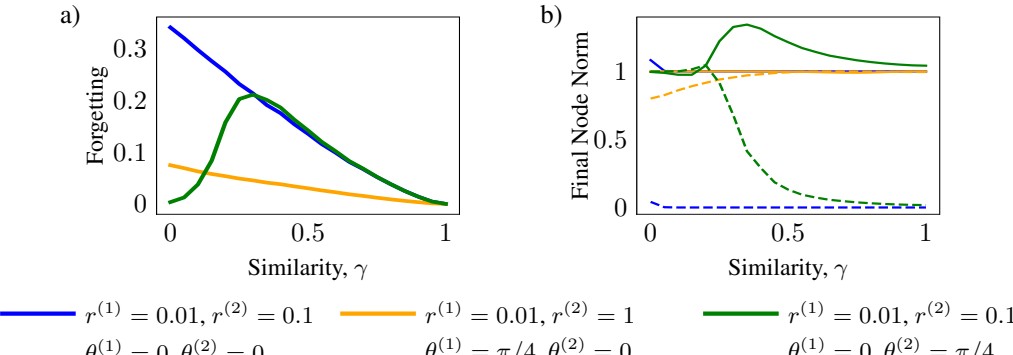

Figure 6: **Initialisation and specialisation properties can influence profile of forgetting vs. similarity.** *(a)* forgetting as a function of task similarity can be both monotonic, shown here for the cases of specialisation after both tasks (blue), and no-specialisation + large, asymmetric second head initialisation (orange); *or* non-monotonic (green, as characterised by Maslow's hammer Lee et al. (2022)). *(b)* the final norm of the two nodes (one solid and one dashed), i.e. at the end of training on both tasks, as a function of task similarity. In the cases that lead to monotonic forgetting, nodes are fully re-used, either because the corresponding new head is initialised large (orange) or because the new head is symmetrically initialised and the nodes continue to represent redundant information during the second task (blue). *Params:* $N = 10000$, $\eta = 1$, $p^* = 1$, $p = 2$, $\sigma_w = 0.001$.

### 4.2.1 SPECIALISATION UNDERLIES MASLOW'S HAMMER.

The phase diagrams in Fig. 5 demonstrate that initialisation can drastically change the type of solutions found by the student after training on one teacher. While this may be inconsequential if the generalisation error remains unaffected, in many cases, the precise nature of the learned representation can significantly impact downstream tasks.

In the worst case scenario, the student undergoes no specialisation during the first task. During the second task there is no notion of the trade-off between node re-use and node activation discussed in Lee et al. (2022); rather the student continues to find a non-specialised solution to the second teacher, effectively fully re-using it's entire representation for the second task. Consequently, the amount of forgetting with respect to the initial task decreases monotonically with task similarity, thereby breaking the inverted U-shaped pattern characteristic of Maslow's hammer that has been observed in various continual learning setups (Ramasesh et al., 2020). This extreme case is illustrated in Fig. 6. Further, *even with* specialisation after the first task, large asymmetric initialisation in the second task readout weights can induce this monotonic relationship, again by pushing the student into re-use rather than activation. In Appendix G, we complement these results with experiments on a task constructed around MNIST and find qualitatively similar results.

In a broader context, a rich diversity of behaviours can emerge, driven by factors such as the initialisation schemes, the scale of weights in the first layer, and the readout heads for both tasks. A glimpse of this behavioural diversity is provided in Appendix F, where we further explore the interaction between these factors and their impact on forgetting in continual learning.

### 4.2.2 SPECIALISATION UNDERLIES EWC.

The findings relating specialisation to forgetting from Sec. 4.2.1 have direct consequences for interference mitigation strategies such as EWC. EWC is a regularisation-based method that computes a measure of "*importance*" for each weight with respect to a task via the Fischer information (Kirkpatrick et al., 2017). Subsequently a squared penalty scaled by this importance is applied to deviation of this weight during learning of future tasks as follows: $\mathcal{L}_{\text{EWC}}(\boldsymbol{W}) = \mathcal{L}(\boldsymbol{W}) + \frac{\xi}{2} \sum_i F_i (W_i - W_i^*)$, where $\boldsymbol{F}$ is the Fischer information matrix, $\xi$ is a regularisation strength parameter, and $\boldsymbol{W}^*$ are the weights at the end of training on the first task.

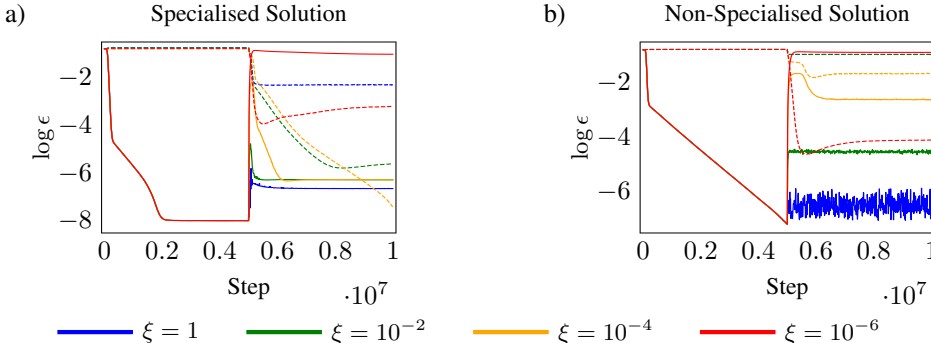

Figure 7: **EWC is strongly reliant on specialisation.** We show the generalisation error in the first (solid line) and second (dashed) task for different EWC regularisation strengths. *(a)* When the student finds a specialised solution to the first task, there is a range of EWC regularisation strength $\xi$ for which the activated units can remain fixed and spare capacity can be used to learn the second task—leading to low generalisation error in both tasks ($\xi = 10^{-2}$, $\xi = 10^{-4}$ perform very well). *(b)* When the student does not specialise in the first task, EWC reduces to an inflexible regulariser that either penalises plasticity everywhere—leading to little forgetting but no further learning (e.g. $\xi = 1$), or does not penalise any plasticity—leading to catastrophic forgetting (e.g. $\xi = 10^{-6}$).

In cases where the network does not specialise, i.e. multiple student nodes learn redundant representations for a given teacher node, the nodes have equal importance. Consequently EWC cannot distinguish between these sets of weights and depending on the regularisation parameter $\lambda$ either lets these nodes move during training on the second task (under-regularises) leading to forgetting, or lets none move (over-regularises) leading to no transfer. We show results illustrating this behaviour in the teacher-student setup in Fig. 7. In particular we show the regime of intermediate task similarity, wherein (Lee et al., 2022) previously argued that EWC should perform better than methods such as replay.

## 5 LIMITATIONS AND PERSPECTIVES

This work operates within simplified frameworks, which–while widely used in the analysis of neural networks–do not fully capture the complexity of modern architectures and real-world data. Our experiments rely on Gaussian input data and simplified input-output relations, which are far from the intricacies of real-world scenarios. A natural next step is to extend our analysis to more realistic generative models, such as the hidden manifold model (Goldt et al., 2020) or the superstatistical generative model (Adomaityte et al., 2023), which offer more structured data distributions and better capture observations from real data experiments. Another promising direction is to complement analytical approaches with numerical experiments on controlled real-world datasets. While this may sacrifice some analytical tractability, it brings us closer to addressing practical challenges. For instance, transfer learning settings, such as those explored in Gerace et al. (2024), provide a useful benchmark for testing our theoretical findings in more complex environments. While we focused on continual learning, other ML domains are affected by specialised representations. An interesting direction concerns the emergence of compositionality. Lepori et al. (2023); Driscoll et al. (2024) reported the emergence of compositional representations in neural networks and theoretical frameworks are now available to investigate the phenomenon (Lee et al., 2024).

While the current work remains theoretical in nature, focusing on simplified models for analytical tractability, a thorough exploration of the practical implications of our findings, particularly in disentangled representation learning, is beyond the scope of this paper. However, we aim to address this in future work by shifting towards a more experimental approach. Specifically, we plan to explore a broader range of network architectures, datasets–such as Car3D (Du et al., 2024) and dSprites (Matthey et al., 2017)–and evaluation metrics—such as SAP (Kumar et al., 2018; Higgins et al., 2017). This future study will allow us to validate our theoretical insights and fully assess their relevance in real-world settings.

ACKNOWLEDGMENTS

This research was funded in whole, or in part, by the Wellcome Trust [216386/Z/19/Z]. For the purpose of Open Access, the author has applied a CC BY public copyright license to any Author Accepted Manuscript version arising from this submission. C.D. and A.S. are supported by the Gatsby Charitable Foundation. A.S. was supported by the Sainsbury Wellcome Centre Core Grant (219627/Z/19/Z) and A.S. is a CIFAR Azrieli Global Scholar in the Learning in Machines & Brains program. S.S.M. was supported by the Wallenberg AI, Autonomous Systems, and Software Program (WASP). D.J. is a Google PhD Fellow mentored by Dr. Gamaleldin F. Elsayed and a Commonwealth Scholar.

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

## A  HYPERBOLIC-LINEAR DYNAMICS

For convenience we will derive the general dynamics here as it requires less notion. Consider a linear network performing a regression task with one hidden layer computing output $\hat{Y} = hWX$ in response to an input batch of data $X$, with $n$ datapoints, and trained to minimize the quadratic loss using gradient descent:

$$L(W, h) = \sum_{i=1}^{n} \frac{1}{2} \|y_i - hW\mathbf{x}_i\|_2^2$$

This gives the learning rules for each layer with learning rate $\eta$ as:

$$\Delta W = \eta n h^T (\Sigma^{yx} - hW\Sigma^x); \qquad \Delta h = \eta n (\Sigma^{yx} - hW\Sigma^x) W^T$$

These equations can be derived for a batch of data using the linearity of expectation, where $\Sigma^x = \mathbb{E}[XX^T]$ is the input correlation matrix and $\Sigma^{yx} = \mathbb{E}[YX^T]$ is the input-output correlation matrix, as follows:

$$\Delta W = \eta \frac{d}{dW} L(W, h) = \eta \frac{d}{dW} \sum_{i=1}^{n} \frac{1}{2} (Y_i - hWX_i)^T (Y_i - hWX_i)$$

$$= \eta \sum_{i=1}^{n} h^T (Y_i - hWX_i) X_i^T = \eta n h^T (\mathbb{E}[Y_i X_i^T] - hW\mathbb{E}[X_i X_i^T])]$$

$$= \eta n h^T (\Sigma^{yx} - hW\Sigma^x)$$

$$\Delta h = \eta \frac{d}{dh} L(W, h) = \eta \frac{d}{dh} \sum_{i=1}^{n} \frac{1}{2} (Y_i - hWX_i)^T (Y_i - hWX_i)$$

$$= \eta \sum_{i=1}^{P} (Y_i - hWX_i)(WX_i)^T = \eta n (\mathbb{E}[Y_i X_i^T] - hW\mathbb{E}[X_i X_i^T])] W^T$$

$$= \eta n (\Sigma^{yx} - hW\Sigma^x) W^T$$

By using a small learning rate $\eta$ and taking the continuous time limit, the mean change in weights is given by:

$$\tau \frac{d}{dt} W = h^T (\Sigma^{yx} - hW\Sigma^x); \qquad \tau \frac{d}{dt} h = (\Sigma^{yx} - hW\Sigma^x) W^T$$

where $\tau = \frac{1}{P\eta}$ is the learning time constant. Here, $t$ measures units of learning epochs. It is helpful to note that since we are using a small learning rate the full batch gradient descent and stochastic gradient descent dynamics will be the same.

Saxe et al. (2019) has shown that the learning dynamics depend on the singular value decomposition of:

$$\Sigma^{yx} = USV^T = \sum_{\alpha=1}^{r_y} \sigma_\alpha u^\alpha v^{\alpha^T}; \qquad \Sigma^x = VDV^T = \sum_{\alpha=1}^{r_x} \delta_\alpha u^\alpha v^{\alpha^T}$$

Here $r_y$ and $r_x$ denote the ranks of the matrices. To solve for the dynamics we require that $\Sigma^{yx}$ and $\Sigma^x$ are mutually diagonalizable such that the right singular vectors $V$ of $\Sigma^{yx}$ are also the singular vectors of $\Sigma^x$. We verify that this is true for the tasks considered in this work and assume it to be true for these derivations. We also assume that the network has at least $r_y$ hidden neurons (the rank of $\Sigma^{yx}$ which determines the number of singular values in the input-output covariance matrix) so that it can learn the desired mapping perfectly. If this is not the case then the model will learn the top $n_h$ singular values of the input-output mapping where $n_h$ is the number of hidden neurons (Saxe et al., 2013). To ease notation for the remainder of this section we will use $n_h$ to denote both the number of hidden neurons and rank of $\Sigma^{yx}$. $S$ and $D$ then are diagonal matrices of the singular values of the input-output correlation and input correlation matrices respectfully.

We now perform a change of variables using the SVD of the dataset statistics. The purpose of this step is to decouple the complex dynamics of the weights of the network, with interacting terms, into multiple one-dimensional systems. Specifically we set:

$$h = U\overline{h}R^T; \qquad W = R\overline{W}V^T$$

where $R$ is an arbitrary orthogonal matrix such that $R^T R = I$. Substituting this into the gradient descent update rules for the parameters above yields:

$$\tau \frac{d}{dt}W = h^T(\Sigma^{yx} - hW\Sigma^x)$$

$$\tau \frac{d}{dt}(R\overline{W}V^T) = R\overline{h}U^T(USV^T - U\overline{h}R^T R\overline{W}V^T VDV^T)$$

$$\tau \frac{d}{dt}(R\overline{W}V^T) = R\overline{h}(SV^T - \overline{h}\overline{W}DV^T)$$

$$\tau \frac{d}{dt}\overline{W} = \overline{h}(S - \overline{h}\overline{W}D)$$

and

$$\tau \frac{d}{dt}h = (\Sigma^{yx} - hW\Sigma^x)W^T$$

$$\tau \frac{d}{dt}(U\overline{h}R^T) = (USV^T - U\overline{h}R^T R\overline{W}V^T VDV^T)V\overline{W}R^T$$

$$\tau \frac{d}{dt}(U\overline{h}R^T) = (US - U\overline{h}\overline{W}D)\overline{W}R^T$$

$$\tau \frac{d}{dt}\overline{h} = \overline{W}(S - \overline{h}\overline{W}D)$$

Here we have used the orthogonality of the singular vectors such that $V^T V = I$ and $U^T U = I$. Importantly, all matrices in the dynamics are now diagonal and represent the decoupling of the network into the modes transmitted from input to the hidden neurons and from hidden to output neurons. In practice we do not initialize the network weights to adhere to this diagonalisation and so it is not guaranteed that the matrices will be diagonal at initialisation. However, empirically it has been found that the network singular values rapidly align to this required configuration (Saxe et al., 2013; 2019).

The derivative then for the full-network input-output mapping can be obtain by using the product rule:

$$\tau \frac{d}{dt}\overline{h}\overline{W} = (\tau \frac{d}{dt}\overline{h})W + h(\tau \frac{d}{dt}W) = \left(\overline{W}(S - \overline{h}\overline{W}D)\right)W + \overline{h}\left(\overline{h}(S - \overline{h}\overline{W}D)\right)$$

$$=\overline{W}^2(S - \overline{hW}D) + \overline{h}^2(S - \overline{hW}D) = \left(\overline{W}^2 + \overline{h}^2\right)(S - \overline{hW}D)$$

This means that at a minimum: $S - \overline{hW}D = 0$ or $\frac{S}{D\overline{W}} = \overline{h}$. This defines a hyperbolic space between $\overline{W}$ and $\overline{h}$. As a result we can use the change of variables: $\overline{W} = \sqrt{\lambda}\sinh\frac{\theta}{2}$ and $\overline{h} = \sqrt{\lambda}\cosh\frac{\theta}{2}$ parametrized by $\theta$.

We note that there is a conserved quantity between the singular values of the weight matrices:

$$\overline{W}^2 - \overline{h}^2 = \left(\sqrt{\lambda}\sinh\frac{\theta}{2}\right)^2 - \left(\sqrt{\lambda}\cosh\frac{\theta}{2}\right)^2 = \lambda$$

This is known as $\lambda$-Balanced weights (Kunin et al., 2024) and for a given initial value for $\lambda$ this quantity will be conserved for all times during training. Aiming to write the network dynamics in terms of this quantity to understand its effect on learning speed and initialisation and with the change of variables to hyperbolic coordinates we begin with:

$$\left(\overline{W}^2 + \overline{h}^2\right)^2 = (\overline{W}^2)^2 + (\overline{h}^2)^2$$
$$= \left(\overline{W}^2 - \overline{h}^2\right)^2 + 4\overline{W}^2\overline{h}^2$$

Substituting this into the network dynamics equation and defining the network singular value as $\omega = \overline{hW}$ we obtain:

$$\tau\frac{d}{dt}\omega = \left(\overline{W}^2 + \overline{h}^2\right)(S - \omega D)$$
$$\tau\frac{d}{dt}\omega = \sqrt{\left((\overline{W}^2 - \overline{h}^2)^2 + 4\overline{W}^2\overline{h}^2\right)}(S - \omega D)$$

Now applying the change of variables to hyperbolic coordinates with $\overline{W} = \sqrt{\lambda}\sinh\frac{\theta}{2}$ and $\overline{h} = \sqrt{\lambda}\cosh\frac{\theta}{2}$ parametrized by $\theta$:

$$\tau\frac{d}{dt}\lambda\cosh\frac{\theta}{2}\sinh\frac{\theta}{2} = \sqrt{\left((\lambda\sinh^2\frac{\theta}{2}) - (\lambda\cosh^2\frac{\theta}{2})\right)^2 + 4\lambda^2(\cosh\frac{\theta}{2}\sinh\frac{\theta}{2})^2}(S - \lambda\cosh\frac{\theta}{2}\sinh\frac{\theta}{2}D)$$

We can then apply the identities: $\cosh\frac{\theta}{2}\sinh\frac{\theta}{2} = \frac{1}{2}\sinh\theta$ and $\lambda\sinh^2\frac{\theta}{2} - \lambda\cosh^2\frac{\theta}{2} = \lambda$:

$$\tau\frac{d}{dt}\frac{\lambda}{2}\sinh(\theta) = \sqrt{\lambda^2 + 4\lambda^2(\frac{1}{2}\sinh(\theta))^2}(S - \frac{\lambda}{2}\sinh(\theta)D)$$
$$\tau\frac{d}{dt}\frac{\lambda}{2}\sinh(\theta) = |\lambda|\cosh(\theta)(S - \frac{\lambda}{2}\sinh(\theta)D)$$

Now applying the derivative on the left:

$$\tau\frac{\lambda}{2}\cosh(\theta)\frac{d}{dt}\theta = |\lambda|\cosh(\theta)(S - \frac{\lambda}{2}\sinh(\theta)D) \tag{11}$$
$$\frac{d}{dt}\theta = \frac{1}{\tau}sgn(\lambda)(2S - \lambda D\sinh(\theta)) \tag{12}$$
$$\tag{13}$$

This is a separable differential equation in $\theta$:

$$\int_{\theta_0}^{\theta_f}\frac{1}{(2S - \lambda D\sinh(\theta))}d\theta = \int_0^t\frac{sgn(\lambda)}{\tau}dt$$

$$\left[\frac{\log\left(\left|2S\tanh\left(\frac{\theta}{2}\right) + \sqrt{4S^2 + \lambda^2 D^2} + \lambda D\right|\right) - \log\left(\left|2S\tanh\left(\frac{\theta}{2}\right) - \sqrt{4S^2 + \lambda^2 D^2} + \lambda D\right|\right)}{\sqrt{4S^2 + \lambda^2 D^2}}\right]_{\theta_0}^{\theta_f} = \frac{sgn(\lambda)}{\tau}t$$

$$\frac{1}{\sqrt{4S^2 + \lambda^2 D^2}} \left[ \log\left( \frac{\left| 2S\tanh\left(\frac{\theta_f}{2}\right) + \sqrt{4S^2 + \lambda^2 D^2} + \lambda D \right|}{\left| 2S\tanh\left(\frac{\theta_f}{2}\right) - \sqrt{4S^2 + \lambda^2 D^2} + \lambda D \right|} \right) \right.$$

$$\left. - \log\left( \frac{\left| 2S\tanh\left(\frac{\theta_0}{2}\right) + \sqrt{4S^2 + \lambda^2 D^2} + \lambda D \right|}{\left| 2S\tanh\left(\frac{\theta_0}{2}\right) - \sqrt{4S^2 + \lambda^2 D^2} + \lambda D \right|} \right) \right] = \frac{sgn(\lambda)}{\tau} t$$

If we let:

$$C = \frac{\left| 2S\tanh\left(\frac{\theta_0}{2}\right) + \sqrt{4S^2 + \lambda^2 D^2} + \lambda D \right|}{\left| 2S\tanh\left(\frac{\theta_0}{2}\right) - \sqrt{4S^2 + \lambda^2 D^2} + \lambda D \right|}; K = \sqrt{4S^2 + \lambda^2 D^2}$$

then:

$$\frac{1}{K} \left[ \log\left( \frac{\left| 2S\tanh\left(\frac{\theta_f}{2}\right) + K + \lambda D \right|}{\left| 2S\tanh\left(\frac{\theta_f}{2}\right) - K + \lambda D \right|} \right) - \log(C) \right] = \frac{sgn(\lambda)}{\tau} t$$

We can further simplify this expression by writing $\theta_f$ in terms of $t$:

$$2S\tanh\left(\frac{\theta_f}{2}\right) + K + \lambda D = C\exp\left(\frac{sgn(\lambda)K}{\tau}t\right)\left(K - 2S\tanh\left(\frac{\theta_f}{2}\right) - \lambda D\right)$$

$$\tanh\left(\frac{\theta_f}{2}\right) = \frac{-K\left(1 - C\exp\left(\frac{sgn(\lambda)K}{\tau}t\right)\right) - \lambda D\left(1 + C\exp\left(\frac{sgn(\lambda)K}{\tau}t\right)\right)}{2S\left(1 + C\exp\left(\frac{sgn(\lambda)K}{\tau}t\right)\right)}$$

$$\theta_f = 2\tanh^{-1}\left( \frac{K\left(C\exp\left(\frac{sgn(\lambda)K}{\tau}t\right) - 1\right) - \lambda D\left(C\exp\left(\frac{sgn(\lambda)K}{\tau}t\right) + 1\right)}{2S\left(C\exp\left(\frac{sgn(\lambda)K}{\tau}t\right) + 1\right)} \right)$$

To obtain the dynamics for the singular value of a mode of the network we use:

$$\omega = \lambda \sinh\frac{\theta}{2}\cosh\frac{\theta}{2}$$
$$= \frac{\lambda}{2}\sinh\theta$$
$$= \frac{\lambda}{2}\sinh\left( 2\tanh^{-1}\left( \frac{K\left(C\exp\left(\frac{sgn(\lambda)K}{\tau}t\right) - 1\right) - \lambda D\left(C\exp\left(\frac{sgn(\lambda)K}{\tau}t\right) + 1\right)}{2S\left(C\exp\left(\frac{sgn(\lambda)K}{\tau}t\right) + 1\right)} \right) \right)$$

In the 1-dimensional case studied in Sec. 3 this equation becomes:

$$\omega(t) = \frac{\lambda}{2}\sinh\left\{ 2\tanh^{-1}\left[ \frac{k\left(c\exp\left(\frac{sgn(\lambda)k}{\tau}t\right) - 1\right) - \lambda d\left(c\exp\left(\frac{sgn(\lambda)k}{\tau}t\right) + 1\right)}{2s\left(c\exp\left(\frac{sgn(\lambda)k}{\tau}t\right) + 1\right)} \right] \right\} \tag{14}$$

with:

$$c = \frac{\left| 2s\tanh\left(\frac{\theta_0}{2}\right) + \sqrt{4s^2 + \lambda^2 d^2} + \lambda d \right|}{\left| 2s\tanh\left(\frac{\theta_0}{2}\right) - \sqrt{4s^2 + \lambda^2 d^2} + \lambda d \right|}; k = \sqrt{4s^2 + \lambda^2 d^2}$$

With the linear network dynamics we can now derive a network's hitting time $(t^*)$ for each mode. Let $v^*$ be a sufficiently small value, using Eq. 14 on the relation $\frac{S}{D} - \omega = v^*$ we obtain

$$\tanh\left(\frac{1}{2}\sinh^{-1}\left(\frac{2S - 2Dv^*}{\lambda D}\right)\right) = \frac{K\left(C\exp\left(\frac{sgn(\lambda)K}{\tau}t\right) - 1\right) - \lambda D\left(C\exp\left(\frac{sgn(\lambda)K}{\tau}t\right) + 1\right)}{2S\left(C\exp\left(\frac{sgn(\lambda)K}{\tau}t\right) + 1\right)}$$

Let $T^* = \tanh\left(\frac{1}{2}\sinh^{-1}\left(\frac{2S - 2Dv^*}{\lambda D}\right)\right)$ then

$$T^* = \frac{K\left(C\exp\left(\frac{sgn(\lambda)K}{\tau}t\right) - 1\right) - \lambda D\left(C\exp\left(\frac{sgn(\lambda)K}{\tau}t\right) + 1\right)}{2S\left(C\exp\left(\frac{sgn(\lambda)K}{\tau}t\right) + 1\right)}$$

$$2ST^*\left(C\exp\left(\frac{sgn(\lambda)K}{\tau}t\right) + 1\right) = K\left(C\exp\left(\frac{sgn(\lambda)K}{\tau}t\right) - 1\right) - \lambda D\left(C\exp\left(\frac{sgn(\lambda)K}{\tau}t\right) + 1\right)$$

$$\exp\left(\frac{sgn(\lambda)K}{\tau}t\right)(2ST^*C - KC + \lambda DC) = -2ST^* - K - \lambda D$$

$$\frac{sgn(\lambda)K}{\tau}t = \log\left(\frac{-2ST^* - K - \lambda D}{2ST^*C - KC + \lambda DC}\right)$$

$$t^* = \frac{\tau}{sgn(\lambda)K}\log\left(\frac{K + 2ST^* + \lambda D}{KC - 2ST^*C - \lambda DC}\right)$$

Similarly we derive the escaping time for each mode with sufficiently small $\hat{v}$ as:

$$\omega = \hat{v}$$

$$\frac{\lambda}{2}\sinh\left(2\tanh^{-1}\left(\frac{K\left(C\exp\left(\frac{sgn(\lambda)K}{\tau}t\right) - 1\right) - \lambda D\left(C\exp\left(\frac{sgn(\lambda)K}{\tau}t\right) + 1\right)}{2S\left(C\exp\left(\frac{sgn(\lambda)K}{\tau}t\right) + 1\right)}\right)\right) = \hat{v}$$

$$\frac{K\left(C\exp\left(\frac{sgn(\lambda)K}{\tau}t\right) - 1\right) - \lambda D\left(C\exp\left(\frac{sgn(\lambda)K}{\tau}t\right) + 1\right)}{2S\left(C\exp\left(\frac{sgn(\lambda)K}{\tau}t\right) + 1\right)} = \tanh\left(\frac{1}{2}\sinh^{-1}\left(\frac{2\hat{v}}{\lambda}\right)\right)$$

Let $\hat{T} = \tanh\left(\frac{1}{2}\sinh^{-1}\left(\frac{2\hat{v}}{\lambda}\right)\right)$ then

$$\hat{T} = \frac{K\left(C\exp\left(\frac{sgn(\lambda)K}{\tau}t\right) - 1\right) - \lambda D\left(C\exp\left(\frac{sgn(\lambda)K}{\tau}t\right) + 1\right)}{2S\left(C\exp\left(\frac{sgn(\lambda)K}{\tau}t\right) + 1\right)}$$

$$2S\hat{T}\left(C\exp\left(\frac{sgn(\lambda)K}{\tau}t\right) + 1\right) = K\left(C\exp\left(\frac{sgn(\lambda)K}{\tau}t\right) - 1\right) - \lambda D\left(C\exp\left(\frac{sgn(\lambda)K}{\tau}t\right) + 1\right)$$

Thus, the escaping time can be summarised as:

$$\hat{t} = \frac{\tau}{sgn(\lambda)K}\log\left(\frac{K + 2S\hat{T} + \lambda D}{KC - 2S\hat{T}C - \lambda DC}\right) \tag{15}$$

with the escaping time constant:

$$\hat{T} = \tanh\left(\frac{1}{2}\sinh^{-1}\left(\frac{2\hat{v}}{\lambda}\right)\right) \tag{16}$$

Similarly the hitting time is summarised as:

$$t^* = \frac{\tau}{sgn(\lambda)K}\log\left(\frac{K + 2ST^* + \lambda D}{KC - 2ST^*C - \lambda DC}\right) \tag{17}$$

with the hitting time constant:

$$T^* = \tanh\left(\frac{1}{2}\sinh^{-1}\left(\frac{2S - 2Dv^*}{\lambda D}\right)\right) \tag{18}$$

Fig. 8 depicts the accuracy of these closed-form equations.

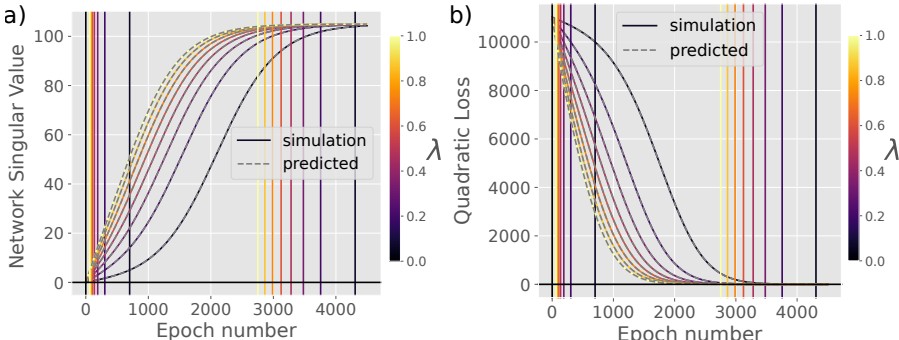

Figure 8: Comparison between the predicted and simulated linear network dynamics. a) depicts the singular value trajectories for varying levels of weight imbalance (depicted in various colours) while gray represents the corresponding predicted trajetectories. b) The same plot for the loss dynamics derived from the singular values. We see exact agreement between the simulations and predictions. Vertical bar depict the escaping time (on left) and hitting time (one right).

## B  METHOD FOR WEIGHT IMBALANCE PHASE PLOT

Given Eqn. 15 and 17 we can discuss our method for construction of the phase plot in Fig. 3d). For each combination of weight imbalanced for the two pathways we aim to find how close the slower pathway is to begin learning at its closest point. We note that merely simulating the full network is not enough as this would merely tell us whether the slower pathway learns something, but with no additional precision. Further, we can also constrain our search space over time by noting that the slower pathway will never be quicker than its initial escaping time. Finally, we share the notation in this section with those in the main text and App. A and omit the notation definitions here. Thus, the algorithm for constructing the phase plot, which we reproduce in Fig. 9, is as follows:

---

**Algorithm 1** An algorithm for constructing Fig. 3d). Hyperparameters used: $S = 105, \Lambda_1 = [0, 100], \Lambda_1 = [0, 20], \hat{\epsilon} = 5.0, \epsilon^* = 1.0, \eta = 1e^{-5}$

---

**Require:** $s, \tau, \hat{\epsilon}, \epsilon^*, \Lambda_1$ and $\Lambda_2$
**Require:** $a(0) > 0$
  $Phase = \mathbf{0}^{|\Lambda_1| \times |\Lambda_2|}$
  **for** $\lambda_1$ in $\Lambda_1$ **do**
    **for** $\lambda_2$ in $\Lambda_2$ **do**
      **if** $\lambda_1 < \lambda_2$ **then**
        break
      **end if**
      $b(0)_1 = \sqrt{(\lambda_1 + a_0^2)}$
      $b(0)_2 = \sqrt{(\lambda_2 + a_0^2)}$
      $\theta_1 = \arcsin^{-1}(2a(0)b(0)_1/\lambda_1)$
      $\theta_2 = \arcsin^{-1}(2a(0)b(0)_2/\lambda_2)$
      $t_1^* = HittingTime(\theta_1, \lambda_1, s, \tau, \epsilon^*)$         ▷ Apply Eqn. 17
      $\omega(t) = Dynamics(\theta_1, \lambda_1, s, \tau, \epsilon^*) \forall t \in [0, t_1^*]$   ▷ Apply Eqn. 3
      **for** $t \in [0, t_1^*]$ **do** $SlowThetas[t+1] = ThetaDeriv(\theta_2, \lambda_2, s - omega(t), \tau, \hat{\epsilon})$
      **end for**         ▷ Numerically integrate coupled slow dynamics using Eqn. 11
      $\hat{t}_2^{coupled} = EscapeTime(SlowThetas, \lambda_2, s - \omega(t), \tau, \hat{\epsilon}) \forall t \in [0, t_1^*]$ ▷ Apply Eqn. 15
      $Phase[\lambda_1, \lambda_2] = \min_t(\hat{t}_2^{coupled}(t))$
    **end for**
  **end for**

---

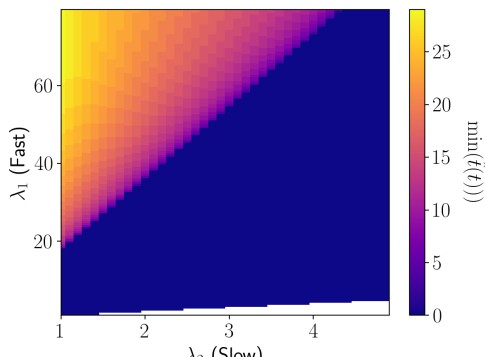

Figure 9: Reproduction of Fig. 3c). A phase diagram representing how pathways with different initial weight imbalances lead to specialisation. The two axis represent the weight imbalance of the two pathways in our broader network ($\lambda_2$ on the x-axis for the slower pathway and $\lambda_1$ on the y-axis for the faster pathway). The colour represents how close the slower pathway is to reaching its escaping time at its closest point throughout training (in $\log$ scale). We see that the more inbalanced the fast pathway relative to the slower pathway, the more likely the network will specialise. The white region represents when the inbalance is equal or reversed.

## C   MEAN-FIELD THEORY OF THE DYNAMICS

As outlined in Sec.4, the key observation for the mean-field analysis is that the main properties of the learning dynamics can be expressed as functions of the order parameters–Eqs. 8. By combining these definitions with the update rules–Eqs. (6, 7)–we can derive closed-form expressions for the evolution of the order parameters, enabling us to track the key observables throughout the training process. In the high-dimensional limit ($d \to \infty$), these discrete update equations converge to ordinary differential equations (ODEs), which can be integrated either numerically or analytically in certain cases (Jain et al., 2024). As is often the case in the statistical physics of disordered systems, this approach was first derived non-rigorously by Saad & Solla (1995b) and Biehl & Schwarze (1995), with later works laying down a mathematical foundation showing concentration of the ODEs (Goldt et al., 2020; Ben Arous et al., 2022).

Following these prescriptions, we obtain the update equations as in Lee et al. (2021). Let us define the pre-activations of the student and task-$t$ teacher given an input $\boldsymbol{x}$ from task $t$ as

$$\lambda_i = \frac{1}{\sqrt{d}}\boldsymbol{W}_i \cdot \boldsymbol{x}, \qquad \rho_i^{(t)} = \frac{1}{\sqrt{d}}\boldsymbol{W}_{T,i}^{(t)} \cdot \boldsymbol{x}, \tag{19}$$

and denote the difference between the teacher and student predictions by $\Delta^{(t)} = \boldsymbol{h}^{(t)} \cdot \phi(\boldsymbol{\lambda}) - \boldsymbol{h}_T^{(t)} \cdot \phi(\boldsymbol{\rho})$. The corresponding ODEs for the order parameters in the limit $d \to \infty$ are given by:

$$\frac{dQ_{ik}}{d\tau} = -\eta h_i^{(t)}\langle\phi'(\lambda_i)\Delta^{(t)}\lambda_k\rangle - \eta h_k^{(t)}\langle\phi'(\lambda_k)\Delta^{(t)}\lambda_i\rangle + \eta^2 h_i^{(t)} h_k^{(t)}\langle\phi'(\lambda_i)\phi'(\lambda_k)(\Delta^{(t)})^2\rangle, \tag{20}$$

$$\frac{dR_{in}^{(t')}}{d\tau} = -\eta h_i^{(t)}\langle\phi'(\lambda_i)\Delta^{(t)}\rho_n^{(t')}\rangle, \tag{21}$$

$$\frac{dh_i^{(t)}}{d\tau} = -\eta\langle\Delta^{(t)}\phi(\lambda_i)\rangle, \tag{22}$$

where $\tau = \text{epoch}/d$ represents continuous time in the high-dimensional limit, and $t, t' \in 1, 2$ denote the task indices. The angular brackets indicate an average over the pre-activations. The pre-activations themselves are centered Gaussian random variables with covariances determined by the order parameters $\boldsymbol{Q}$, $\boldsymbol{R}^{(t)}$, and $\boldsymbol{T}$.

These averages can be computed analytically for certain activation functions. For instance, in the case of a rescaled error function introduced in the main text (Saad & Solla, 1995b; Biehl &

Schwarze, 1995), the relevant averages are given by:

$$\langle \phi(\beta)\phi(\gamma)\rangle = \frac{1}{\pi}\arcsin\left(\frac{\Sigma_{12}}{\sqrt{(1+\Sigma_{11})(1+\Sigma_{22})}}\right), \tag{23}$$

$$\langle \phi'(\zeta)\beta\phi(\gamma)\rangle = \frac{2\Sigma_{23}(1+\Sigma_{11})-2\Sigma_{12}\Sigma_{13}}{\sqrt{\Lambda_3}(1+\Sigma_{11})}, \tag{24}$$

$$\langle \phi'(\zeta)\phi'(\iota)\phi(\beta)\phi(\gamma)\rangle = \frac{4}{\pi^2\sqrt{\Lambda_4}}\arcsin\left(\frac{\Lambda_0}{\sqrt{\Lambda_1\Lambda_2}}\right), \tag{25}$$

where the Greek letters represent arbitrary pre-activations with covariance matrix $\boldsymbol{\Sigma}$, and the auxiliary quantities $\Lambda_i$ are given by:

$$\Lambda_0 = \Lambda_4\Sigma_{34} - \Sigma_{23}\Sigma_{24}(1+\Sigma_{11}) - \Sigma_{13}\Sigma_{14}(1+\Sigma_{22}) + \Sigma_{12}\Sigma_{13}\Sigma_{24} + \Sigma_{12}\Sigma_{14}\Sigma_{23}, \tag{26}$$

$$\Lambda_1 = \Lambda_4(1+\Sigma_{33}) - \Sigma_{23}^2(1+\Sigma_{11}) - \Sigma_{13}^2(1+\Sigma_{22}) + 2\Sigma_{12}\Sigma_{13}\Sigma_{23}, \tag{27}$$

$$\Lambda_2 = \Lambda_4(1+\Sigma_{44}) - \Sigma_{24}^2(1+\Sigma_{11}) - \Sigma_{14}^2(1+\Sigma_{22}) + 2\Sigma_{12}\Sigma_{14}\Sigma_{24}, \tag{28}$$

$$\Lambda_3 = (1+\Sigma_{11})(1+\Sigma_{33}) - \Sigma_{13}^2. \tag{29}$$

These expressions provide a comprehensive analytical framework for tracking the dynamics of the student network and the evolution of specialisation across training.

## D   DISENTENGLEMENT

We conduct our experiments using open-source frameworks Locatello et al. (2019); Abdi et al. (2019). Specifically, we implement a beta-VAE with the "DeepGaussianLinear" architecture for the decoder and "DeepLinear" for the encoder. We modify the Xavier initialisation where the weights of the linear layers will have values sampled from $U(-a, a)$ with

$$a = \text{gain} \times \sqrt{\frac{6}{\text{fan\_in} + \text{fan\_out}}}$$

We vary the gain between 0.3 and 3 and run each experiment over 4 seeds. All network parameters are set to their default values as provided by the respective open-source frameworks. We run the experiments for 20 Epochs and 157499 iterations.

These experiment illustrate the impact of initialisation on network specialisation. Although the scope of these experiments is limited, they provide preliminary validation of our theoretical framework in more realistic contexts. We advocate for further investigation into alternative initialisation schemes with varying levels of balance. Moreover, we highlight the need for future research to extend these experiments by considering a wider variety of datasets (Car3D Du et al. (2024), dSprites Matthey et al. (2017),), network architectures (Conv,Linear), initialisation strategies ( Gaussian Xavier Initalisation) and different metric (SAP Kumar et al. (2018); Higgins et al. (2017),) to fully explore the implications of our findings.

**DCI Disentanglement** Eastwood & Williams (2018) define three key properties of learned representations: Disentanglement, Completeness, and Informativeness. To assess these, they calculate the importance of each dimension of the representation in predicting a factor of variation. This can be done using models like Lasso or Random Forest classifiers. Disentanglement is computed by subtracting the entropy of the probability that a representation dimension predicts a factor, weighted by its relative importance. Completeness is similarly measured, focusing on how well a factor is captured by the dimensions. Informativeness is evaluated as the prediction error of the factors. We use the implementation in Locatello et al. (2019). In this implementation, we sample 10,000 training and 5,000 test points, then use gradient-boosted trees from Scikit-learn to obtain feature importance weights. These weights form an importance matrix, with rows representing factors and columns representing dimensions. Disentanglement is calculated by normalizing the columns of this matrix, subtracting the entropy from 1 for each column, and then weighting by each dimension's relative importance.

## E   ADDITIONAL ENTROPY PHASE DIAGRAMS

In Fig. 5 we showed phase diagrams of the aggregate entropy as a function of initialisation parameters, for both ReLU and sigmoidal networks. In Fig. 10 below, we show additional plots with the individual entropy terms ($H_u$ defined over the unit activations, and $H_h$ defined over the head weights).

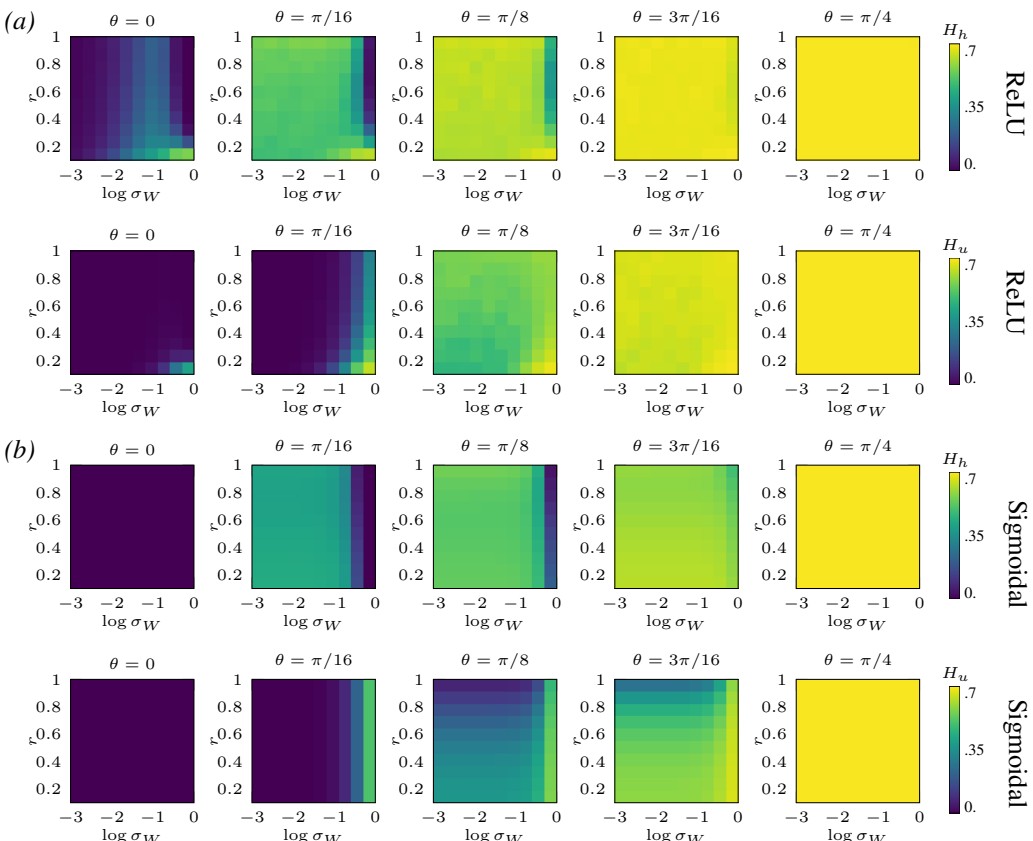

Figure 10: **Additional Phase Diagrams.** Here we show the equivalent phase diagrams from Fig. 5 for entropy measures over the unit activations and head weights.

## F   DIVERSITY OF FORGETTING CURVES

In Fig. 6 we show that initialisation can lead to a variety of specialisation profiles in contrast with what observe in previously.

## G   FORGETTING CURVES IN MNIST

In order to support the findings presented in Sec. 4.2.1, we turn now to a continual learning task constructed around the MNIST dataset . This dataset has previously been adapted to continual learning benchmarks e.g. most famously in the permuted MNIST task . Here we construct a slightly different continual learning task to encode a notion of task similarity.

We begin by considering only one half of the 10 class MNIST dataset, such that we are left with only data in the first 5 classes. Our first task in the sequence of two tasks consists simply of classifying these 5 digits. Our second task is also to classify 5 digits and ranges from classifying the same 5 digits (maximum task similarity) to classifying 5 new digits, i.e. those that were discarded to construct the first task (orthogonal tasks—minimum task similarity). In a 10 class dataset like

MNIST this gives us only a very coarse grip on task similarity, but this is enough to robustly elicit behaviour analogous to what we observe in the toy teacher-student models.

We use a two-layer, multi-head, feed-forward architecture with sigmoidal activations to mirror the models used in the teacher-student setup. The hidden dimension of our networks needs to be larger to properly learn the classification task; we therefore lose the elegance and control afforded by the polar coordinate initialisations of Sec. 4.2.1 to vary entropy and scale of initialisations. The method we use to generalise this notion is to interpolate between two initialisations: a high entropy initialisation (e.g. a uniform distribution), and a relatively low entopy initialisation (e.g. Normal or Laplace distribution). It is straightforward (but important) to ensure the scale of the samples generated is consistent across this interpolation.

In Fig. 12, we show forgetting profiles for three different initialisation schemes (analogous to those shown in Fig. 6) for the continual MNIST task described above. It is clear that in the case of low entropy and specialisation in the first task along with high entropy second head initialisation, we get behaviour characteristic of Maslow's hammer. However when we initialise the second head with low entropy, we recover the monotonic relationships found in the equivalent initialisations from the toy models. At this stage these are primarily qualitative results, i.e. we are comparing the shapes of these forgetting profiles and not the relative magnitudes or detailed forgetting metrics.

## H  SPARSE AUTO ENCODER EXPERIMENT

To further verify the applicability of the linear network theory presented in Sec. 3 we experimentally verify a prediction from the theory. Sec. 3 finds that networks which are initialised with larger hidden-to-output weights compared to the input-to-hidden weights will have a specialisation benefit. As we mention in the main text, the notion of specialisation in this work is very similar to activation sparsity. As a consequence, we predict that by leveraging an output heavy initialisation scheme we can improve the sparsity of an autoencoder.

We conduct the following experiment in two phases: *Phase 1:* We train a standard VAE (similar to Sec. 3.2) on MNIST which was initialised with small weights to ensure the network is in the feature learning regime (Geiger et al., 2020) (we sample from a Gaussian with standard deviation $0.001$). Importantly, the latent dimension of this VAE is smaller than the input and forms an entangled latent space. *Phase 2:* In a similar manner to the recent approach on the Claude line of Large Language Models (Templeton, 2024), we train a sparse autoencoder (SAE) from the latent space of the VAE, with the aim of improving the sparsity and disentanglement of the latent space. In our experiments the VAE has 16 hidden neurons. These 16 neurons become the input (and output) to the SAE. The SAE then projects this up to a latent space of dimension $2048$ which has a ReLU activation function. For our baseline, we train the SAE with the typical L2 reconstruction loss and *L1 regularisation on the hidden activity*. For our model, we train in exactly the same manner, except we *do not use the L1 regularisation on the hidden activity*. Thus, for our model there is no explicit pressure on the autoencoder to embed representations sparsely. For ease we will refer to this model as an implicit Sparse Autoencoder (iSAE). We repeat this process with varying degrees of initialisation imbalance and track the sparsity of the SAE and iSAE. Denoting the hidden layer activity of the networks for the entire MNIST dataset as $H$ we define an indicator function in Eqn. 30 for a single neuron responding to a single datapoint:

$$\mathbf{1}(H_{ij}) =: \begin{cases} 1, & H_{ij} > 0 \\ 0, & \text{otherwise} \end{cases} \tag{30}$$

We calculate the sparsity across the dataset as the average number of datapoints the hidden neurons respond to, over the 60000 datapoints:

$$\frac{1}{2048} \sum_{i=1}^{2048} \sum_{j=1}^{60000} \mathbf{1}(H_{ij}) \tag{31}$$

To initialise the layers of the iSAE and SAE we define an imbalance parameter $\upsilon$ (note that this is not the same hyper-parameter as the $\lambda$ notation employed in the main text and is defined purely for

practicality in this experiment). The encoder weights are initialised by sampling from a Gaussian with standard deviation $\sigma = 0.001\frac{1}{\upsilon}$. The decoder weights are sampled from a Gaussian with standard deviation $\sigma = 0.001\upsilon$. Thus, as $\upsilon$ increases the decoder is initialised with increasingly large weights compared to the decoder.

The results of this experiment are shown in Fig. 13. We see clearly from these results that as the initialisation imbalance is pushed towards the hidden-to-output weights such that they are larger than the input-to-hidden weights, then the sparsity of the iSAE latent space improves dramatically. This corresponds to a positive $\lambda$-balance in the theoretical results and, thus, our empirical and theoretical results are consistent. This is in spite of there only being an implicit bias towards sparsity. Conversely the SAE with explicit sparsity regularization does not change in response to varying degrees of initialisation imbalance. Importantly, this provides empirical support for the findings from the linear network dynamics and verifies our prediction resulting from this theory.

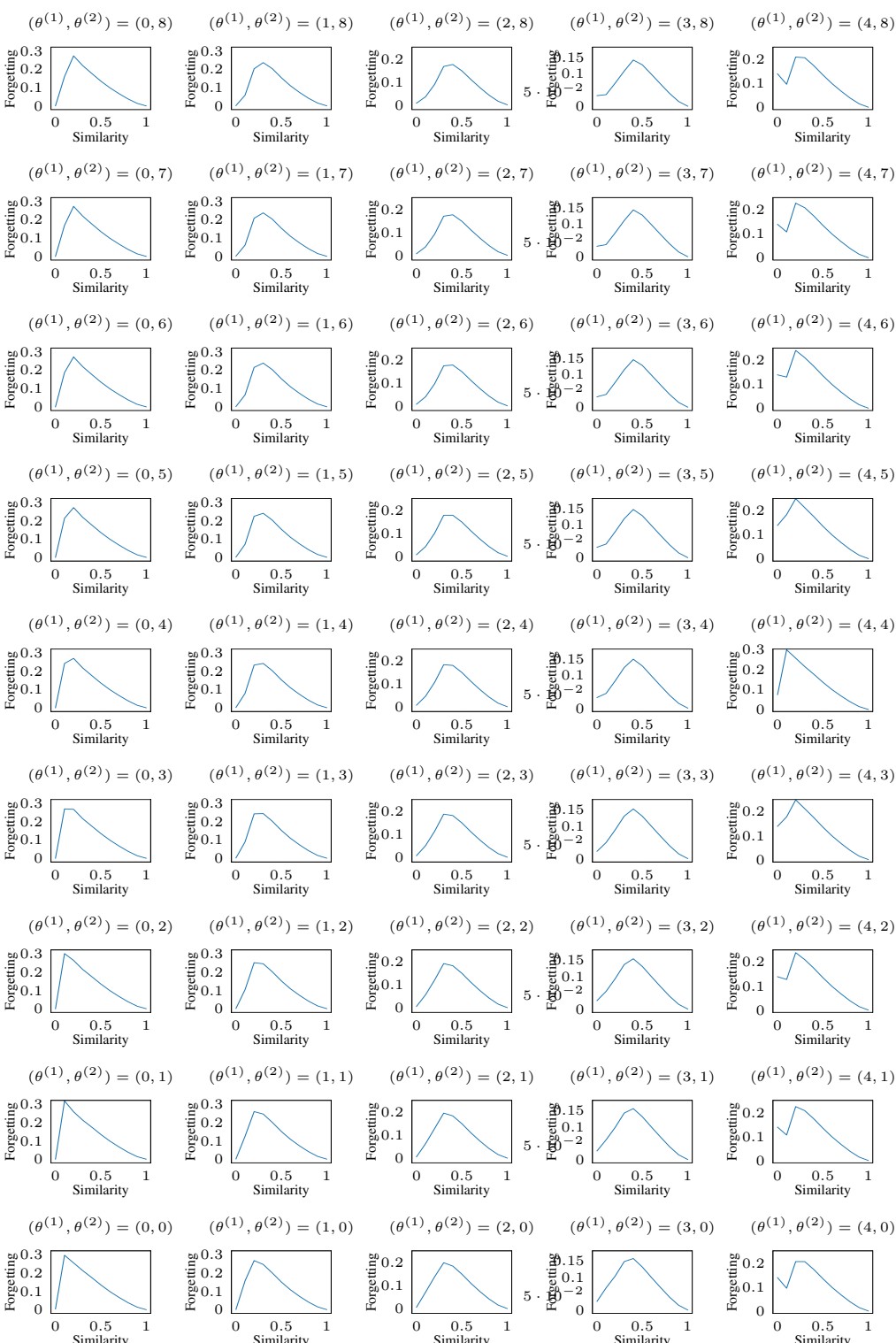

Figure 11: **Initialisation can lead to a diversity of specialisation dynamics and a diversity of relationships between forgetting and task similarity.** $R, \sigma_W$ fixed, $\theta^{(1)}, \theta^{(2)}$ measured in increments of $\pi/16$. Scaled error function, $P^* = 1, P = 1$.

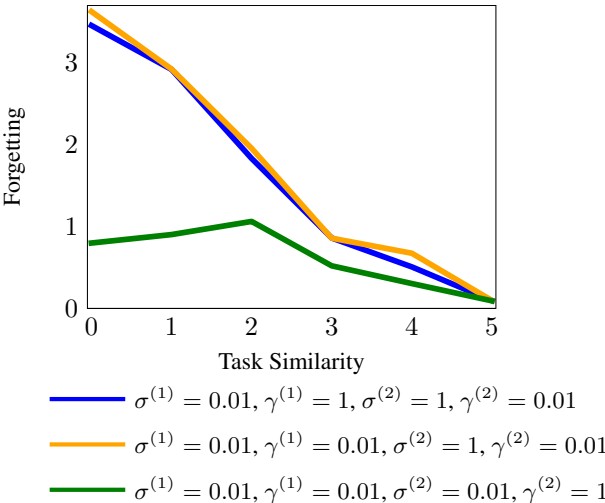

Figure 12: **Forgetting profiles on MNIST continual learning problem.** Forgetting vs. task similarity on a continual learning task using the MNIST dataset. Here similarity is defined as the the number of classes that are the same in a 5-way classification problem from the first task to the second, i.e. 0 corresponds to 5 new classes and 5 corresponds to the same 5 classes. The green line is achieved by initialising with low entropy and small weights in the first head followed by low entropy and small weights in the second, while the blue and orange lines have low entropy second head initialisations with high and low entropy initialisations in the first head respectively. These forgetting profiles (in terms of their monotonocity patterns) qualitatively match those observed in the theoretical toy problems discussed in Sec. 4.2.1 (see Fig. 6). Note $\sigma^{(i)}$ denotes the scale of the $i^{\text{th}}$ head initialisation (equivalent to $r$ in Fig. 6) and $\gamma^{(i)}$ the relative entropy (plays similar role to $\theta$ in Fig. 6).

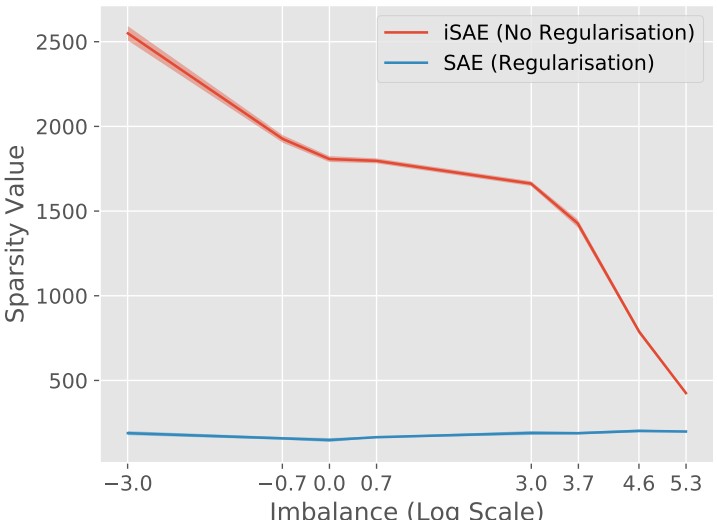

Figure 13: **Implicit regularisation from initialisation imbalance:** We track the sparsity of the iSAE and SAE for varying degrees of initialisation imbalance (x-axis). The imbalance on the x-axis depicts the natural log of the imbalance parameter ($v$). Thus, 0.0 depicts balanced initialisation typically used in practice. The y-axis depicts the corresponding sparsity calculated using Eqn. 31. Clearly, as the imbalance increases the sparsity of the iSAE decreases (which is consistent with the findings from the linear network theory of Sec. 3), while the SAE does not respond due to its explicit regularisation. Results depict the average over ten runs with two standard deviations on either side of the mean.

