# OpenReview forum: "A Theory of Initialisation's Impact on Specialisation"
_ICLR.cc/2025/Conference — ICLR 2025 Poster_

### Official Review · Reviewer_i4VX · 2024-10-28

**Soundness:** 3
**Presentation:** 3
**Contribution:** 2
**Rating:** 6
**Confidence:** 1

**Summary:**

The paper presents an analysis of specialization in neural networks without nonlinearities, with focus on the determinants of specialization and its influence on forgetting.

The analytical methods of the paper fall outside my expertise hence I cannot give a meaningful evaluation of the paper as a whole as they make up the majority of the argument. The below ratings and comments are limited to a superficial evaluation of the main points and should not be regarded as reliable.

**Strengths:**

The analysis and discussion about the relevance of specialziation to forgetting, as well as the determinants of specialization, is an interesting area on inquiry.
The analysis is provided with a rigorous mathematical framework.

**Weaknesses:**

The analysis is performed on linear networks, while nonlinearity is the defining feature of NNs. Authors are clear about that and apparently it is not an unprecedented way to conduct a similar analysis, but this would render the conclusions of the analysis questionable nonetheless. I would suggest the authors to include a more detailed discussion regarding what sort of guarantees can be given when transferring conclusions obtained from linear networks to conventional, nonlinear ones.

**Questions:**

Please see weaknesses point above.

---

> ### Author Response · Authors · 2024-11-21
> **Detailed reply to reviewer i4VX**
>
> We thank the reviewer for their feedback. We will address highlighted weaknesses in detail, with the aim of strengthening your confidence in the importance of our research.
>
> **Performed on linear networks.** While Sec.3 considers the theory of Deep Linear Networks (Saxe et al. 2013), Sec.4 presents theoretical results with a complementary framework suitable to generic non-linearity (Goldt et al. 2019). In particular, Fig.1 shows results with ReLU and Sigmoidal. The inclusion of two established theoretical paradigms is a strength of this work (and, to our knowledge, a very uncommon trait). Moreover, we include the two theories in a manner that mitigates their respective weaknesses. To elaborate, the teacher-student framework used in Section 4 is capable of describing nonlinear dynamics with larger datasets and network sizes. However, the exact mechanism of specialisation is less apparent as it considers the correlation of hidden neurons in the dynamics. The deep linear network dynamics, however, describe the race between two neurons to account for variance in the data and clearly reflect how an imbalance in the weights connected to individual neurons will benefit specialisation. Noting the consistency in the ultimate findings of the two theories and having aimed to mitigate their respective weaknesses, we are then able to be far more confident with the result. Finally, we note that the experiments conducted in Section 3.2 are for nonlinear networks in a naturalistic setting, and are motivated by established experimental designs in the machine learning literature (Locatello et al. 2019). Thus, these experiments serve to demonstrate the generality of our findings alongside the linear dynamics theory of Section 3 and the non-linear stat-physics theory of Section 4. In conclusion, we thank the reviewer for highlighting the need to discuss the structure of this paper in more detail, as we have carefully considered how the various perspectives and methodologies combine to provide a contribution greater than the sum of their individual parts. We will try to be clearer at the end of the introduction, where we discuss the structure of the paper and the various design decisions. We hope this addresses the reviewer’s noted weakness.

---

> > ### Comment · Reviewer_i4VX · 2024-11-25
> >
> > It seems that the weakness I mentioned stems from a misunderstanding in my part and it's good that the authors are will make changes to make the text clearer so that other readers won't have the same misunderstanding.
> > I will keep my score as a placeholder since, as iterated before, it shouldn't be regarded as a reliable evaluation of the work due to the topic lying outside my field of expertise.

---

> > > ### Author Response · Authors · 2024-11-25
> > > **Response to Reviewer i4VX by Authors**
> > >
> > > We thank the reviewer again for their consideration of our work and will certainly use the review to guide how we improve the clarity of the paper. We appreciate the reviewer's perspective which will help us ensure that our work is clear to the broader machine learning community present at ICLR.

---

### Official Review · Reviewer_yUsa · 2024-11-04

**Soundness:** 2
**Presentation:** 1
**Contribution:** 3
**Rating:** 6
**Confidence:** 3

**Summary:**

The paper investigates how weight initialization, specifically imbalance, affect the arising of specialization in the network representation and the subsequent consequences on forgetting wrt. task similarity.

**Strengths:**

Originality: Although specialization and continual learning have been widely studied in the context of modularity, this paper attempts to take a more thorough investigation from the perspective of weight imbalance at initialization which is interesting and an important direction.

Quality: Figure 3&4 provide good insights on how imbalance leads to specialization/disentanglement both theoretically and experimentally.

Clarity: Notations are clear and maths are largely clear (except Eq. 9).

Significance: I think weight imbalance is an interesting and important angle for continual learning thus the paper has great potential.

**Weaknesses:**

Quality: 1) As mentioned by the authors, simplified setting is a limitation of the paper: all theoretical results are on single-hidden layer networks (linear for learning dynamic, nonlinear for student-teacher); All experiments including the ones on continual learning are on toy examples except Figure 4. 2) I find the overall main message unclear - is specialization good or bad for continual learning?  (see question)

Clarity:  The current flow of the paper can be problematic, it required significant back-and-forth for me to understand the messages (example see questions).  The manuscript seems incomplete, for example the caption for Figure 4.

Significance: Apart from the other mentioned comments, one key aspect the paper could gain on its significance is to make more clear conclusions (see questions).

**Questions:**

a. Conflicting results? - Figure 6 orange is no specialization and with lowest forgetting (good?) but Figure 7 trying to say no specialization is not good? Is specialization good or bad for continual learning?

b. Can the authors give any explicit conclusion on weight imbalance and their impact on continual learning, apart from initialization has an effect? This seems to be the most important message of the paper but I can't find any statement in the text. Eq 9 gives the generalization error but was not explained.

c. Fig 6 is very confusing. Two factors: layer-wise imbalance (r) vs. pathway imbalance (theta). But they are both changing across scenarios; without controls, I don't see how conclusions can be drawn.

d. Figure 6 blue - why is it 'specialization after the first task' since theta=pi/4 and according to Figure 5 that's yellow = no specialization?

e. Unclear conclusion: For example, Sec 4.3 on continual learning, its conclusion only mentions the concerned factors without making clear their consequences on CL so standing more like an introduction than a conclusion "In a broader context, a rich diversity of behaviours can emerge, driven by factors such as the initialisation schemes, the scale of weights in the first layer, and the readout heads for both tasks."

f. Could the authors somehow materialize the concept of 'specialization' in Figure 2? I could vaguely see how two pathways of different lambda could lead to specialization as one simply wins the race but it was after much back-and-forth comprehension.

g. Figure 1b main message = activation function is subsidiary to weight imbalance? If so, it seems to be disconnected from the rest of the paper and seems like better belonging to the appendix than as a killer figure.

h. Flow problem: Q and R in Figure 1 are not really introduced until in Sec 4.1 but even then they are only defined without explained (l352).

---

> ### Author Response · Authors · 2024-11-21
> **Detailed reply to reviewer yUsa**
>
> We thank the reviewer for their consideration of our work. We find their questions very useful to improve the clarity of the message. One of the challenges of framing our contribution is in combining insights from two theoretical approaches that are usually not presented together. We believe that the comments and the ones of the other reviewers will help us polish the message.
>
> **The manuscript seems incomplete, for example the caption for Figure 4.** Thank you for pointing this out, we unfortunately missed the abrupt end in one of our submission revisions. We will fix it.
>
> **Is specialisation good or bad for continual learning? And use of Toy Examples**
> Our theoretical results demonstrate how imbalances and entropy in initialisation impact specialisation. Based on these results, we explore how this specialisation shapes the forgetting profile, concluding that the initialisation for Task 1 significantly affects the forgetting profile for Task 2. This paper doesn't try to label one scenario as good or bad per se. Specifically, Sec. 4 highlights two different forgetting profiles established in the continual learning literature. Namely, the Maslow's Hammer profile, observed first in Ramasesh et al. (2021), and the monotonic forgetting profile, more typically assumed and observed in Goodfellow et al. (2014). The main aim of Sec. 4 is to show that which profile a network follows depends on how nodes trained during the first task are reused for a second. In this work, we focus on how the latent representations learned during the first task impacts forgetting. In the case of small initialisation of the second readout head, if a distributed representation is used, then node reuse will occur when a second task is learned. However, if specialised representations are learned in the first task, then subsequent tasks will have excess capacity to learn new information and salient reusable features to build off of. Thus, it is important to understand the factors influencing specialisation in the first task.  We believe that the only way to isolate and study specialisation is in toy models, as they remove most of the confounding effects that we observe in real datasets and more complex architectures. They can also benefit from mathematical analysis, which would be impossible otherwise. We will comment on it in the revision.
>
>
> **Fig 6.** First we must apologise that there are a couple of typos in the legend + caption of Figure 6, which we will correct in the revised version. The keys for the blue and orange line have been switched by mistake. The caption has then been muddled as a result; the caption should, therefore, say "(a) forgetting as a function of task similarity can be both monotonic, shown here for the cases of specialisation after both tasks (blue), and no specialisation + large, asymmetric second head initialisation (orange); or non-monotonic (green, as characterised by Maslow's hammer)". Hopefully, that will clear up the consistency with Fig 5 and our conclusions as a whole. More generally to your comment about controls in Fig 6, the aim of this figure is to give examples of the forgetting profiles that can emerge from different regimes of learning. The aim is not necessarily to draw quantitative conclusions but rather to highlight how different initialisation schemes can lead to different specialisation behaviours and different downstream continual learning dynamics. We will try to make the nature and scope of this figure clearer in our revisions.
>
> **Unclear conclusion.** We will try to sharpen the sentence highlighted. Indeed it is aimed to be a summarising / concluding sentence. In part, the current breadth of scope of this passage reflects the fact that the behaviours we are analysing are complex and multi-faceted, and there are various effects we have identified (i.e. across weight imbalance, entropy of initialisations within each layer, etc.). We attempt to elucidate this variety when discussing the results individually (e.g. in the figures) and provide a higher-level perspective in the concluding paragraph.
>
> **The concept of 'specialisation' in Figure 2?** In this case, we wanted to discuss specialisation along the lines of the lottery ticket hypothesis (Frankle, Carbin 2018) or Neural Race (Saxe et al. 2022), where a network can process information along different pathways, and we wanted to see if there was a dominant one or all pathways were used simultaneously. The results of this investigation are in Fig.3, where we can see a clear change of behaviour depending on initialisation. We will include this link in the caption of Figure 2 as we believe it provides an intuitive link to an established concept in the literature. We thank the reviewer for the suggestion. Please also see the response to CSfH, where we discuss in more detail comparisons to other definitions of specialisation in the literature.

---

> ### Author Response · Authors · 2024-11-21
> **Detailed reply to reviewer yUsa**
>
> **Figure 1b main message = activation function is subsidiary to weight imbalance?** Fig.1b is primarily a motivational figure and also grounds some claims we make in previous work in the literature. In particular, we contrast with previous results that observed that sigmoidal networks specialise while ReLU networks do not (Goldt et al. 2019). In contrast, we show here that with certain initialisations, ReLU networks can also specialise, and sigmoidal networks do not always specialise. Our point in this figure is not that activations are subsidiary to weight imbalance (or vice versa) but rather that initialisation schemes also play an important role in specialisation behaviour for online SGD, which so far has been under-appreciated by the theory community. Thus, our results provide new perspectives and insights that contradict conventional wisdom.
>
> **Flow problem: Q and R in Figure 1 are not really introduced until in Sec 4.1.** We feel it will be difficult for narrative flow to introduce the mathematical definition and details there (it will require quite some space) but we can improve the text by saying that the notion of specialisation requires sparser Q and R matrices pointing to the section with additional details.
>
> We also addressed the reviewer's minor comments.

---

### Official Review · Reviewer_CSfH · 2024-11-04

**Soundness:** 3
**Presentation:** 3
**Contribution:** 3
**Rating:** 6
**Confidence:** 2

**Summary:**

The paper studies how the initialization of a network can affect its ability to “specialize” where representations are localized and disentangled. The paper discusses a simple student-teacher framework with extensions to disentangled attribute learning and continual learning. I think the paper looks solid but it is highly likely that I didn’t understand the paper.

**Strengths:**

- The paper is informative on the theoretical setup and its implication in empirical experiments such as disentangled learning and continual learning.
- The conclusion about network initialization is insightful.
- The figures in the paper are nicely organized.

**Weaknesses:**

- Clarity: I found the paper not very clear. First it builds off from some prior knowledge on “specialization” and its definition of specialization is “one pathway finish learning (reach it’s hitting time t ∗) before the other begins learning (reaches it’s escaping time tˆ)”. However, “hitting” and “escaping” are two new concepts without definition. And why does the leading time matter for specialization? Can’t a MoE like architecture also specialize? In other words, the two neurons could simultaneously specialize into two different aspects of the input data, where the lead time of one neuron does not matter. Perhaps this is because I am very new to the literature, but without properly understanding the concept of specialization, I cannot have a confident judgment on the paper.
- Although the paper is theoretical, the main paper does not have any formal theorem/claim to explain the relation between initialization and specialization.
- The paper lacks a core claim. The first part attributed specialization to the “imbalance” in initialization (I also didn’t understand why h-ww^T is imbalance). The second part focuses on how “gain” of initialization affects disentanglement. And lastly, in continual learning, initialization was mentioned but unclear which part of initialization can cause the failure of continual learning. Basically, it seems that the whole paper is talking about initialization but the actual mechanism is quite different in each part and lacks a unifying explanation.

**Questions:**

See weaknesses

---

> ### Author Response · Authors · 2024-11-21
> **Detailed reply to reviewer CSfH**
>
> We thank the reviewer for their feedback. One of the challenges of framing our contribution is that we wanted to combine insights from two theoretical approaches that are usually not presented together. Thank you for helping us identify criticalities in our explanation, we will improve it following the reviewers suggestions. Below we address each of the concerns you highlighted in detail, with the aim of strengthening your confidence in the importance of our research.
>
> **Clarity concerns.**
> - We kindly direct the reviewer towards Lines 161 and 176, where we note, "When will one pathway finish learning (reaches its hitting time $t^*$) before the other begins learning (reaches its escaping time $\hat{t}$)". We recognise that while these are the definitions of escaping and hitting time, stating in this manner hinders clarity. To be specific here, hitting time is how long it takes a pathway to reach its final converged value. Escaping time is how long it takes the pathway to begin learning. Thus, considering the singular value of a pathway as defined in Equation 3, hitting time is the time $t$ such that $\omega(\infty) - \omega(t) < \delta$ for some small $\delta \in \mathbb{R}$. Similarly, the escaping time is the time $t$ such that $\omega(t) > \delta$ for some small $\delta \in \mathbb{R}$. We will add these more concrete definitions to a subsequent draft and thank the reviewer for their assistance with the clarity of our work.
> - We kindly direct the reviewer towards Lines 147 to 152 of Section 3. Here, we note that the notion of specialisation typically employed in works such as MoE is slightly different from the notion of specialisation in statistics physics literature. In the statistical physics paradigm of theory, a single neuron needs to account for an aspect of the input data (see, e.g. Goldt et al. 2019) where the "aspect" corresponds to the singular vectors being learned. In the more architectural approach, typical of MoEs, what matters is that some subset of the network is responsive to an intuitive aspect of the input data. In this case, the "intuitive aspect" corresponds to an interpretable feature - forming an expert module on that aspect (see e.g. Andreas et al.  2016; Jarvis et al. 2023). We will emphasise this subtlety more in a subsequent draft and thank the reviewer again for their assistance with the clarity of our work.
> - For the second portion of the question on there being "two different aspects of the input data" we agree with the reviewer and note that the analysis of Section 3 considers a rank-1 task without loss of generality. Since we use a rank-1 task, there is only one aspect of the input to specialise to, and one hidden neuron can learn the task. However, if there were multiple, then the same consideration would hold for all of the aspects. Importantly, as we perform the derivations in terms of the singular value decomposition of the dataset (see Equation 1), these aspects do not compete. Thus, the singular vectors would depict what the "aspect" being learned is, and the singular value over time (which we conduct the dynamics derivation of) corresponds to the learning of the aspect. Since the singular vectors are orthogonal, the aspects cannot interfere. The interpretability (we can interpret the singular vectors) and tractability (dynamics of learning) are the two greatest strengths of the linear dynamics paradigm. Thus, if we were to scale the dataset and network correspondingly, individual neurons in the hidden layer would race to explain a rank-1 aspect as they do in our analysis. Once a neuron aligns to explain one aspect then it cannot impact learning of other aspects and the remaining neurons will race to explain the remaining aspects, and so on.

---

> ### Author Response · Authors · 2024-11-21
> **Detailed reply to reviewer CSfH**
>
> **Does not have any formal theorem/claim.**
> In this paper we presented a theoretical physics approach to machine learning. Contrary to mathematics and classical theoretical computer science approaches, physics-based theory papers don't follow the theorem/proof structure. There are several important differences; the most salient one is that some of these methods are based on heuristic arguments whose results have been proved rigorously only in some specific cases but for which a general theorem is not available (e.g. the replica method for integration - see Gabrié et al. (2023) [5] for a review). In these cases, the theoretical argument is often followed by a numerical confirmation of the theory; some recent examples are [1-3], and the space includes a large variety of topics from CNNs and transformer representations to curriculum learning. The physics-based approach often proposes a modelling approach to understand the problem in simplified, solvable settings. We also include a series of books and review papers where additional information on this approach can be found [4-7]
> .
> - [1] Cagnetta, Francesco, et al. "How deep neural networks learn compositional data: The random hierarchy model." Physical Review X 14.3 (2024): 031001.
> - [2] Mannelli, Stefano Sarao, et al. "Tilting the Odds at the Lottery: the Interplay of Overparameterisation and Curricula in Neural Networks." arXiv preprint arXiv:2406.01589 (2024).
> - [3] Garnier-Brun, Jérôme, et al. "How transformers learn structured data: insights from hierarchical filtering." arXiv preprint arXiv:2408.15138 (2024).
> - [4] Andreas Engel and Christian Van den Broeck. Statistical mechanics of learning Cambridge University Press, 2001
> - [5] Marylou Gabrié, Surya Ganguli, Carlo Lucibello, and Riccardo Zecchina. Neural networks: from the perceptron to deep nets. ArXiv preprint, abs/2304.06636, 2023. URL https://arxiv.org/abs/2304.06636
> - [6] Zdeborová, Lenka, and Florent Krzakala. "Statistical physics of inference: Thresholds and algorithms." Advances in Physics 65.5 (2016): 453-552.
> - [7] Charbonneau, Patrick, et al., eds. Spin Glass Theory and Far Beyond: Replica Symmetry Breaking after 40 Years. World Scientific, 2023.
>
>
> **The paper lacks a core claim.**  We agree with the reviewer that the clarity of the core claim could be improved. We will address this by improving the introduction and problem definition. To elaborate here: our claim is that initialisation has a fundamental importance in achieving a specialised solution. Indeed, our paper starts from the claim that specialisation is severely impacted by initialisation. Thus, Sec. 3 tries to understand in more detail "what" aspect of the initialisation scheme leads to a specialised solution. Initialisation is not a single variable to analyse, and our investigation has to consider concepts like "weight imbalance" and "initialisation entropy". In Sec. 4, we then demonstrate the importance of understanding this effect for downstream tasks, including continual learning.
>
> **Continual learning, initialisation was mentioned but unclear which part of initialisation can cause the failure of continual learning.** Continual learning is an interesting case where specialisation matters. Section 4 aims to use the new insights on initialisation's role in specialisation from Sec 3. to explain two established forgetting profiles empirically observed in the literature. Namely, the Maslow's Hammer profile, observed first in Ramasesh et al. (2021), and the monotonic forgetting profile, more typically assumed and observed in Goodfellow et al. (2014). Thus, specialised solutions lead to minimal catastrophic forgetting due to it, leading to the Maslow's Hammer profile (Lee et al. 2022). Thus, our analysis shows that not only can initialisation lead to bad forgetting, but it can also disrupt standard mitigation strategies like Elastic Weight Consolidation EWC (Kirkpatrick et al. 2017).

---

> ### Author Response · Authors · 2024-11-21
> **Detailed reply to reviewer CSfH**
>
> **Lacks a unifying explanation.**
>  In this paper, we highlight how two aspects of initialisation play a crucial role in building specialised representations. Specifically, we show across different settings that both imbalance and entropy in the initialisation interact and lead to varying degrees of specialisation. In Figure 1, within the student-teacher framework, we demonstrate the influence of these two factors while also noting the role of the activation function for continual learning. In Section 3, using a linear network, we delve into a mechanistic explanation of the effect of entropy in the output layer compared to the input layer. Then, in our disentangled experiments aimed to support the theoretical results, we leverage the fact that, in expectation, standard network initialisations (e.g., LeCun or He initialisation) yield $\lambda$-balanced weights, where $\lambda$ corresponds directly to the gain we control (Dominé et al., 2024). Together, these results demonstrate how imbalances and entropy in initialisation impact specialisation. We next explore how this specialisation shapes the forgetting profile, concluding that the initialisation for Task 1 significantly affects the forgetting profile for Task 2. Specifically, Sec. 4 highlights two different forgetting profiles established in the literature. Namely, the Maslow's Hammer profile, observed first in Ramasesh et al. (2021), and the monotonic forgetting profile more typically assumed and observed in Goodfellow et al. (2014). The main aim of Sec. 4 is to show that which profile a network follows depends on how nodes trained during the first task are reused for a second. In this work we focus on how the latent representations learned during the first task impacts forgetting. In the case of small initialisation of the second readout head, if a distributed representation is used, then node reuse will occur when a second task is learned. However, if specialised representations are learned in the first task, then subsequent tasks will have excess capacity to learn new information and salient reusable features to build off of. Thus, it is important to understand the factors influencing specialisation in the first task.

---

> > ### Comment · Reviewer_CSfH · 2024-11-25
> >
> > I thank the authors for their response. It helps me understand the paper better. I think the paper has positive values and I maintain the score of 6 (leaning towards acceptance).

---

> > > ### Author Response · Authors · 2024-11-25
> > > **Response to Reviewer CSfH by Authors**
> > >
> > > We thank the reviewer again for their consideration of our work and overall positive evaluation of the paper. We are committed to making the most of the discussion period. Thus, if the reviewer has any further suggestions which would improve their evaluation of the paper or strengthen their confidence in the merit of our work, we would gladly aim to incorporate these.

---

### Official Review · Reviewer_SPqY · 2024-11-10

**Soundness:** 4
**Presentation:** 3
**Contribution:** 3
**Rating:** 6
**Confidence:** 3

**Summary:**

- The authors introduce a theoretical framework that demonstrates how initialization controls whether networks develop specialized or shared representations
- This work contains a rigorous mathematical analysis demonstrating that weight imbalance between layers drives specialization
- This paper presents a novel analysis of how initialization affects continual learning methods, particularly demonstrating the dependency of EWC on specialization

**Strengths:**

- This work takes a new perspective on the class catastrophic forgetting problem and provides theoretical justification for why initialization is important
- I believe continual learning researchers will be interested in this approach and in building on this work

**Weaknesses:**

- The primary weakness is the lack of empirical experiments. I would like to see different initialization methods compared on several simple continual learning or class incremental learning benchmarks. Overall, I think this work still has value as a theory paper.

**Questions:**

How would your initialization schemes scale to modern architectures with skip connections?

---

> ### Author Response · Authors · 2024-11-21
> **Detailed reply to reviewer SPqY**
>
> We thank the reviewer for the time and effort dedicated to reviewing our manuscript. We will address each of the concerns you highlighted in detail, with the aim of strengthening your confidence in the importance of our research.
>
> **Lack of empirical experiments.** We thank the reviewer for pointing out their concern. We intended this work to be mainly a theoretical contribution that characterises the impact of initialisation on specialisation and continual learning. Section 4 aims to use the new insights on initialisation’s role in specialisation to explain two established forgetting profiles empirically observed in the literature. Namely, the Maslow’s Hammer profile, observed empirically first in Ramasesh et al. (2021), and the monotonic forgetting profile, more typically assumed and observed in Goodfellow et al. (2014). Hence, our work aims to explain empirical results theoretically, and there is merit to our approach without empirics (see e.g. Lee et al. 2021). Nevertheless, we agree with the reviewer on the importance of empirical validations beyond the assumptions of the model. Indeed, in the paper, we provided some results on the disentangled representation learning problem in Sec.3.2. The aim of this section is to empirically supplement the theoretical results of Sec. 3.1.
>
> **Initialisation schemes scale to modern architectures with skip connections.** We agree with the reviewer that skip connections are an interesting consideration and can offer some preliminary discussion on the topic. The two theoretical paradigms employed in this work, deep linear networks with hyperbolic dynamics and the student-teacher dynamics, are both restricted to architectures with a single hidden layer, rendering an analysis of skip connections impossible. However, we know that **in the balanced weight regime** ($\lambda = 0$ in this work), networks learn slower with the addition of more hidden layers (Saxe et al. 2019). Secondly, pathways through a neural network race to “explain” the variance in the dataset (Saxe et al. 2022) - this “Neural Race” is foundational to the analysis of Section 3. Combining these perspectives, we can come to a clear hypothesis: A skip connection jumping a particular set of hidden layers will race the pathway going through these hidden layers. As it is shallower it will have a learning speed advantage over the deeper pathway. At this point, specialisation will depend on the behaviour of the skip connection alone (as the slower pathway has not begun learning). If the pathway with the skip connection has a single hidden layer of neurons, then we return to the analysis in Section 3. Otherwise, if we extrapolate our findings (and acknowledge that this alone should be verified in future work), we would speculate that the larger the weights in a layer of a network $W_i$, the more likely the weights in the earlier layers $W_j$ for $j < i$ will be to specialise (while noting that there are likely many factors at play beyond initialisation). Finally, if there is also a sufficient imbalance in the initialisation between the deeper and skip-connection pathways (with the deeper pathway being initialised larger), then it is possible that the deeper pathway will account for some or all of the variance. If that is all, then the skip connection has no bearing, and we can again extrapolate from our findings here. If the pathways are balanced in their learning speed, then variance could be shared, and the analysis becomes extremely intricate. The most immediate question would be, how does one compare neurons in different pathways and at different depths of the network? Clearly, the definitions and analysis of this case become highly intricate and require thorough investigation, which we hope you appreciate is beyond the scope of this paper. Thus, we consider this future work and hope that the analysis offered here will benefit such a study.

---

### Author Response · Authors · 2024-11-21
**General response to the review**

Thank you to the reviewers for your detailed and thoughtful feedback. We really appreciate the time and effort you put into reviewing our manuscript—your input has been key to improving its quality. We’ve carefully addressed each of your comments and included linked responses for any overlapping points. To summarise the key updates we will make to the revised manuscript:

**Clarity**
We agree with the reviewer’s comments regarding clarity. A key challenge in presenting our contributions lies in integrating insights from two theoretical approaches that are typically not combined, yet are complementary in that they can address each other’s limitations. We have revised all sections highlighted by the reviewers for clarification and corrected all identified typographical errors. We hope these updates will make the paper’s message clearer, as detailed in the individual responses.

**Message**
This paper aims to show that the initialisation has a fundamental importance in achieving a specialised or unspecialised solution. Our paper starts by showing that specialisation is severely impacted by initialisation. Fig.1b serves as a motivation figure and contrasts with results from the literature (Goldt et al. 2019) that concluded that sigmoidal networks specialise while ReLU networks do not. However, we show that ReLU networks can also specialise (and sigmoidal networks do not always specialise) depending on weight initialisations. The rest of the paper then tries to understand in more detail “what” aspects of the initialisation scheme lead to a specialised solution. Initialisation is not a single variable to analyse, and our investigation has to consider concepts like weight imbalance and entropy; as a result, there is not necessarily one single punchline result. Finally, we demonstrate the importance of understanding this effect for downstream tasks, including continual learning. Specifically, our analysis shows that not only can initialisation lead to bad forgetting, but it can also disrupt standard mitigation strategies like Elastic Weight Consolidation EWC (Kirkpatrick et al. 2017).

**Experiments**
We intended this work to be mainly a theoretical contribution that characterises the impact of initialisation on specialisation, beginning in the linear network regime in Sec. 3.1. Subsequently, Section 4 aims to use the new insights on initialisation’s role in specialisation to explain two established forgetting profiles empirically observed in the literature. Namely, the Maslow’s Hammer profile, observed empirically first in Ramasesh et al. (2021), and the monotonic forgetting profile, more typically assumed and observed in Goodfellow et al. (2014). Hence, our work aims to explain empirical results theoretically, and there is merit to our approach without empirics (see, e.g. Lee et al. 2021). Nevertheless, we agree with reviewers on the importance of empirical validations beyond the assumptions of simple models. Indeed, in the paper, we provided some results in the disentangled representation learning problem, Sec.3.2. The aim of this section is to empirically supplement the theoretical results of Sec. 3.1.

We hope these revisions enhance the clarity and impact of our work and that you find our manuscript a valuable contribution to the field. We look forward to any further feedback you may have!

---

### Author Response · Authors · 2024-11-24
**Discussion period ending soon**

Dear Area Chair and Reviewers,

We have submitted detailed responses to all the reviewers' concerns, along with a general response summarizing the changes made. With only one day remaining in the discussion period, we kindly ask for confirmation on whether our replies have addressed the reviewers' concerns, and, if so, to consider adjusting your score accordingly.

Your feedback is crucial to improving the quality of the paper, and we would greatly appreciate your engagement before the deadline.

Thank you for your time and consideration.

Best regards,
The Authors

---

### Author Response · Authors · 2024-11-27
**Revised Draft from Reviewer Feedback**

We thank the reviewers again for their feedback. An updated version of the manuscript has been uploaded where we *addressed the clarity concerns* of the reviewers and *added two novel experiments on MNIST whose results support our theories*. Here we provide a summary of the findings of the two novel experiments:
1. To support the findings on the forgetting profiles in the student-teacher setting, we turn to a more realistic continual learning task constructed around the MNIST dataset. We show that the forgetting profiles (in terms of their monotonocity or Maslow’s Hammer patterns) qualitatively match those observed in the theoretical toy problems discussed in the paper.
2. The second experiment supports our finding that a large imbalance between hidden-to-output and input-to-hidden weights leads to higher specialization, derived from the linear network theory. As noted in the main text, the concept of specialization in this study closely aligns with activation sparsity. We compared two kinds of autoencoders trained on MNIST: a first. where sparsity was enhanced using the weight imbalance; a second autoencoder where sparsity was induced via a regularisation approach recently used in the large language model literature [1].

Furthermore, to the best of judgement we have provided detailed responses to all reviewers’ concerns and adjusted the revised draft accordingly. For easy comparison, the main changes in the revised draft have been written in blue. In particular Reviewers **SPqY** and **yUsa** asked explicitly for more experiments to support the theory. We believe the additional experiments should address these concerns.

We are committed to making the most of the discussion period. Thus, if the reviewers have any further questions which would strengthen their evaluation or confidence in the merit of our work, we would gladly aim to answer these. Thank you for your time and consideration.

[1] Templeton, Adly. Scaling monosemanticity: Extracting interpretable features from Claude 3 sonnet. Anthropic, 2024.

---

### Meta-Review · Area_Chair_2hWC · 2024-12-20

**Metareview:**

This paper looks at how initialisation affects a network's ability to 'specialise' its representation, adopting a theoretical approach, and with an application in continual learning (the impact on forgetting due to task similarity).

Reviewers agree that the theory is useful and has interesting implications, and the conclusions are interesting. However, reviewers also agree upon a key limitation: most reviewers found it hard to understand the paper, commenting on the paper's overall clarity and message. During the author rebuttal and subsequent discussion, some of the reviewers agree that the paper is now clearer, and I agree. I also think that the authors can still improve readability to focus on the key messages of the paper, and this will further improve the reach of this paper. Some examples have been provided by reviewers, such as Reviewer yUsa:
- "For example, in the unchanged abstract, instead of saying 'assessing the implications on EWC', maybe say something like 'specialization by weight imbalance is beneficial for EWC to reduce forgetting'. The same applies to defining Q and R. I would encourage the authors to spend more efforts on improving the readability of their paper so that their important message on weight imbalance / specialization / sparsity could actually be delivered to the broader community."

**Additional Comments On Reviewer Discussion:**

I think the additional experiments provided by the authors during the rebuttal period help, but are of smaller impact than the clarity. I found some of the comments provided by authors during rebuttal (such as in the 'general response to the review') to be better than in the paper, and recommend the authors to work on changing the writing to further reflect the points brought up by reviewers (into the abstract and introduction too).

Reviewer i4VX could not check the analytical methods of the paper (outside their expertise), but it is still useful to see that they had difficulties understanding key takeaways and the understanding the discussion (shared by other reviewers).

Reviewer SPqY asked for empirical experiments as a weakness, and authors added some during the rebuttal. I agree that this is not of primary importance / is not the main contribution of the paper.

Reviewer yUsa found that after authors fixed a mistake in Fig 6, that it helped, and improved their score to 6, like the other reviewers.

---

### Decision · Program_Chairs · 2025-01-22

Accept (Poster)